# Predicting Label Distribution from Ternary Labels

**Yunan Lu, Xiuyi Jia**[*]
School of Computer Science and Engineering
Nanjing University of Science and Technology, Nanjing 210094, China
{luyn, jiaxy}@njust.edu.cn

## Abstract

Label distribution learning is a powerful learning paradigm to deal with label polysemy and has been widely applied in many practical tasks. A significant obstacle to the effective utilization of label distribution is the substantial expenses of accurate quantifying the label distributions. To tackle this challenge, label enhancement methods automatically infer label distributions from more easily accessible multi-label data based on binary annotations. However, the binary annotation of multi-label data requires experts to accurately assess whether each label can describe the instance, which may diminish the annotating efficiency and heighten the risk of erroneous annotation since the relationship between the label and the instance is unclear in many practical scenarios. Therefore, we propose to predict label distribution from ternary labels, allowing experts to annotate labels in a three-way annotation scheme. They can annotate the label as "0" indicating "uncertain relevant" if it is difficult to definitively determine whether the label can describe the instance, in addition to the binary annotation of "1" indicating "definitely relevant" and "−1" indicating "definitely irrelevant". Both the theoretical and methodological studies are conducted for the proposed learning paradigm. In the theoretical part, we conduct a quantitative comparison of approximation error between ternary and binary labels to elucidate the superiority of ternary labels over binary labels. In the methodological part, we propose a Categorical distribution with monotonicity and orderliness to model the mapping from label description degrees to ternary labels, which can serve as a loss function or as a probability distribution, allowing most existing label enhancement methods to be adapted to our task. Finally, we experimentally demonstrate the effectiveness of our proposal.

## 1 Introduction

LDL (Label Distribution Learning) [2] is an effective learning paradigm for addressing label polysemy (i.e., the cases where an instance can be described by multiple labels). Distinct from multi-label learning [22], which assign a binary-valued vector to each instance, LDL assigns each instance a real-valued vector, akin to descrete probability distributions, to represent the description degree of each label to the instance. Label distributions provide fine-grained information about label polysemy, and thus have been applied to many practical tasks, such as sentiment analysis [8, 30, 35], facial age estimation [1, 4, 24] and movie rating prediction [3].

A fundamental bottleneck hindering LDL is the difficulty in acquiring ground-truth label distributions, as the accurate quantification of these distributions can be prohibitively expensive. Therefore, LE (Label Enhancement) [29] is proposed to automatically infer label distributions from the more easily accessible multi-label data. Multi-label data is based on binary annotations (i.e., utilizing binary values $\pm 1$ to annotate each label) which demand that experts accurately identify whether each label

---

[*]Corresponding author

38th Conference on Neural Information Processing Systems (NeurIPS 2024).

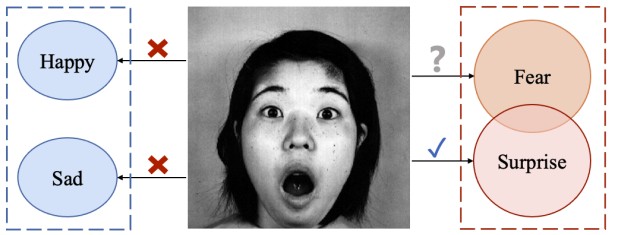

| Label | Annotation |
|---|---|
| Happy | irrelevant $(-1)$ |
| Sad | irrelevant $(-1)$ |
| Surprise | relevant $(+1)$ |
| Fear | uncertain $(?)$ |

Figure 1: An annotation from JAFFE dataset [18].

can describe the instance. However, accurate identification is challenging in real-world tasks due to the prevalence of ambiguous instances and labels which provoke uncertain binary associations between the instances and the labels. For example, the facial image in Figure 1 can be definitely (negatively or positively) associated to "happy", "sad", and "surprise", whereas the association with "fear" is uncertain, since it is not clear whether the emotion comes from being stimulated by something frightening. If experts are coerced into providing a definitive annotation in uncertain cases, not only does it diminish the efficiency of annotating, but it also heightens the risk of erroneous annotation.

Therefore, we propose to predict label distribution from ternary labels. Inspired by the philosophies of three-way decision or three-world thinking [31], where decision-makers are provided with the flexibility to delay judgment when the available information is inadequate to support a determined decision, ternary labels take values from $\{-1, 0, 1\}$, where "$\pm 1$" indicates whether the label can describe the instance, and "0" denotes that the relationship between the label and the instance is uncertain. For the proposed learning paradigm, we conduct theoretical and methodological studies. In the theoretical part, we first quantify the errors of approximating the ground-truth label description vector [2] from ternary labels and binary labels, respectively. Further, we conduct a quantitative comparison of approximation error between ternary and binary labels to elucidate the superiority of ternary labels. In the methodological part, we propose CateMO distribution (Categorical distribution with Monotonicity and Orderliness) to model the mapping from label description degrees to ternary labels, which can serve as the probability distribution for generating ternary labels or as the loss function for measuring the inconsistency between the ternary labels and label distributions, allowing most existing LE methods to be adapted to our task. Specifically, we first analyze the rules governing the generation of ternary labels from label description degrees, and formalize them as the assumptions about the probabilistic monotonicity and orderliness of ternary labels. Further, we derive the probability mass function of CateMO to ensure probabilistic monotonicity and orderliness. Finally, we create two comparison algorithms and evaluate the prediction performance on three real-world datasets. Experimental results unequivocally affirm the superiority of ternary labels. Our contributions can be summarized as follows:

- We propose to predict label distribution from ternary labels, which not only enhances the annotation accuracy but also significantly reduces the annotating cost when contrasted with the traditional binary annotating methods.

- We rigorously analyze the error of approximating the ground-truth label description degrees by ternary and binary labels, respectively, which provides a quantitative elucidation of the superior performance of the ternary label.

- We propose the CateMO distribution specifically designed to capture the mapping from label description degrees to ternary labels, which is theoretically constructed to maintain the monotonicity and ordinality of the probabilities associated with ternary labels.

## 2  Related Work

To address the challenge of obtaining accurate ground-truth label distributions, LE (Label Enhancement) [29] is proposed as a method to automatically infer label distributions from more accessible

---

[2]In general, the elements of a label distribution are assumed to be non-negative and sum to 1. For simplicity, we disregard the restriction that the sum is 1 in this paper, and refer to the unnormalized label distribution as label description vector. The element of a label description vector is called label description degree.

multi-label data, including binary labels [29] and multi-label rankings [13, 15]. Existing works on LE can be broadly categorized into discriminative LE methods and generative LE methods. Discriminative LE directly treats the label distribution as conditional probabilities of the labels given feature observations, and subsequently mine additional information to estimate the conditional probabilities. Generative LE, on the other hand, focus on describing the generation process of observations in a principled manner and uncovering the underlying patterns of the observed data.

Current discriminative LE methods generally strive to optimize both $\text{Dist}(z, y)$ and $\Omega(z)$, where $\text{Dist}(z, y)$ measures the inconsistency between the label description degrees $z$ and the more accessible labels $y$ (including binary labels and multi-label rankings), and $\Omega(z)$ is the regularization terms based on various sources of information. Typically, $\text{Dist}(z, y)$ is modeled using MSE (Mean Squared Error), and the information for $\Omega(z)$ includes instance relationships and label correlations. For instance, several algorithms [5, 10, 20, 21, 23, 33, 34] assume that the instance manifolds based on features are similar to the instance manifolds based on label distributions. Additionally, several algorithms [11, 26, 29, 32] operate under the assumption that instances with similar feature vectors exhibit similarity in their label distributions. To incorporate label correlations, some algorithms [17, 25] regularize the label distributions using the Graph Laplacian operators of label correlation graphs. LEPNLR [6], on the other hand, enhances the labels by preserving the ranking relation of labels. Existing generative LE methods generally decompose the joint distribution of complete data as $p(y|z)$, $p(\cdots|z)$, and $p(z)$, where $p(y|z)$ models the relationship between $y$ and $z$, and $p(\cdots|z)$ captures the relationships between $z$ and other observed variables. Current works primarily focus on modeling $p(y|z)$ and $p(\cdots|z)$. For instance, LEVI [27, 28] and GLEMR [12] model $p(y|z)$ as a Gaussian distribution and a ranking-preserved distribution, respectively. GLERB [16] and LEIC [14] model $p(\cdots|z)$ by incorporating instance relationships and label correlations into generation processes.

Furthermore, similar to ternary labels, the binary labels with missing values [17] are formally represented by $\{0, \pm 1\}$. However, it should be emphasised that they inherently differ in the following aspects. First, they differ in the origin of the labels with value of $0$, i.e., the missing labels in the literature [17] and the uncertain label in ternary labels. The missing labels stem from the "absence", i.e., the association between the label and the corresponding instance is undocumented or unannotated rather than undeterminable. By contrast, the uncertain labels stem from the "uncertainty", i.e., it is difficult for experts to definitively determine whether the label is relevant to the corresponding instance. Second, they differ in the range of the underlying label description degree. The description degree of the missing labels to the corresponding instance can take any value in the interval $[0, 1]$, since the missing labels may be relevant labels, irrelevant labels, or uncertain labels. By contrast, the description degree of the uncertain labels to the corresponding instance take values in a small sub-interval of $[0, 1]$, which is much tigher than the range of the description degrees of missing labels.

## 3 Quantitative analysis

### 3.1 Preliminary

Given an instance with feature vector of $x$, the label description vector of the instance is denoted by $z$ whose $m$-th element $z_m \in [0, 1]$ indicates the description degree of the $m$-th label to the instance $x$. If an instance is annotated with binary labels, the label description vector will be degenerately expressed as a vector of binary values $b \in \{-1, 1\}^M$ whose $m$-th element $b_m$ denotes whether the $m$-th label can describe the instance or not. If an instance is annotated with ternary labels, the label description vector will be degenerately expressed as a vector of ternary values $s \in \{-1, 0, 1\}^M$ where $s_m = 1$ and $s_m = -1$ denotes that the $m$-th label can describe and cannot describe the instance, respectively, and $s_m = 0$ denotes that the association between the $m$-th label and the instance $x$ is uncertain.

### 3.2 Approximation error analysis

To quantify the advantages of the ternary label over the binary label, we leverage EAE (Expected Approximation Error) [13, 15] which measures the error of approximating the true label description degree by a more accessible label such as binary label, multi-label ranking, or ternary label. The definition of EAE is formalized as follows:

**Definition 1** *Given a label with the true label description degree $z \in \mathcal{I}$, if the label is annotated by a reduced label with the approximate label description degree $\hat{z} \in \hat{\mathcal{I}}$, then the expected approximation*

*error of the reduced label, i.e., $\psi(\hat{\mathcal{I}}, \mathcal{I})$, is quantified as:*

$$\psi(\hat{\mathcal{I}}, \mathcal{I}) = \int_{z \in \mathcal{I}} \int_{\hat{z} \in \hat{\mathcal{I}}} \frac{1}{V\hat{V}} (z - \hat{z})^2 \mathrm{d}\hat{z}\mathrm{d}z, \quad V = \int_{z \in \mathcal{I}} \mathrm{d}z, \quad \hat{V} = \int_{\hat{z} \in \hat{\mathcal{I}}} \mathrm{d}\hat{z}. \tag{1}$$

According to Definition 1, it can be seen that EAE depends on $\mathcal{I}$ and $\hat{\mathcal{I}}$, so below we propose the following assumptions to determine $\mathcal{I}, \hat{\mathcal{I}}$ and their relationships.

**Assumption 1** *As stated in the introduction, a label in practical cases can be positive (i.e., the label can describe the instance), negative (i.e., the label cannot describe the instance), and uncertain (i.e., the association between the label and the instance is uncertain). If a label is positive, then the true label description degree $z \in (\kappa, 1]$. If the label is negative, then the true label description degree $z \in [0, \tau)$, where $\tau \in [0, \kappa]$. If the label is uncertain, then the true label description degree $z \in [\tau, \kappa]$. That is, $z \in \mathcal{I}_s$,*

$$\mathcal{I}_s = \big\{ z : \big(s = 1 \wedge z \in (\kappa, 1]\big) \vee \big(s = -1 \wedge z \in [0, \tau)\big) \vee \big(s = 0 \wedge z \in [\tau, \kappa]\big) \big\}. \tag{2}$$

*If a label is annotated by a ternary label $\hat{s}$, then we have the $\hat{s}$-based label description degree $\hat{z} \in \mathcal{I}_{\hat{s}}$,*

$$\mathcal{I}_{\hat{s}} = \big\{ \hat{z} : \big(\hat{s} = 1 \wedge \hat{z} \in (\hat{\kappa}, 1]\big) \vee \big(\hat{s} = -1 \wedge \hat{z} \in [0, \hat{\tau})\big) \vee \big(\hat{s} = 0 \wedge \hat{z} \in [\hat{\tau}, \hat{\kappa}]\big) \big\}, \tag{3}$$

*where $0 \leq \hat{\tau} \leq \hat{\kappa} \leq 1$ are the predefined parameters as approximations to $\tau$ and $\kappa$, respectively, since $\tau$ and $\kappa$ are unavailable from the annotation results.*

*If a label is annotated by a binary label $\hat{b}$, then we have the $\hat{b}$-based label description degree $\hat{z} \in \mathcal{I}_{\hat{b}}$,*

$$\mathcal{I}_{\hat{b}} = \big\{ \hat{z} : \big(\hat{b} = -1 \wedge \hat{z} \in [0, \hat{\xi})\big) \vee \big(\hat{b} = 1 \wedge \hat{z} \in [\hat{\xi}, 1]\big) \big\}, \tag{4}$$

*where $\hat{\xi} \in [\hat{\tau}, \hat{\kappa}]$ must hold, otherwise there exists some label description degree corresponding to "$\hat{b} = 1 \wedge \hat{s} = -1$" or "$\hat{b} = -1 \wedge \hat{s} = 1$", which is semantically contradictory.*

**Assumption 2** *Since a label can be positive, negative or uncertain in practical cases, we discuss the relationships between the true label and the annotation results in each of the three cases below.*

*If a positive or negative label (i.e., $s = \pm 1$) is annotated by a ternary value $\hat{s}$ or a binary value $\hat{b}$, then we have $\hat{s} = \hat{b} = s$.*

*If an uncertain label (i.e., $s = 0$) is annotated by a ternary value $\hat{s}$ or a binary value $\hat{b}$, then we have $\hat{s} = s$, $p(\hat{b} = -1|s = 0) = \rho$, and $p(\hat{b} = 1|s = 0) = 1 - \rho$, where $0 \leq \rho \leq 1$.*

Next, we give the analytical form of EAE of the binary label and the ternary label.

**Theorem 1** *Given a label equiprobably being positive, negative, and uncertain, if the label is annotated by a ternary value, then we have the expected approximation error:*

$$\mathbb{E}_{\hat{s}, s}[\psi(\mathcal{I}_{\hat{s}}, \mathcal{I}_s)] = \frac{2}{9}(\tau + \kappa)^2 + \frac{2}{9}(\hat{\tau} + \hat{\kappa})^2 - \frac{1}{6}(\hat{\tau}\kappa + \hat{\kappa}\tau) - \frac{1}{3}(\hat{\tau} + \kappa)(\hat{\kappa} + \tau) + \frac{1}{18}(1 - \kappa - \hat{\kappa}). \tag{5}$$

*If the label is annotated by a binary value, then we have the expected approximation error:*

$$\mathbb{E}_{\hat{b}, s}[\psi(\mathcal{I}_{\hat{b}}, \mathcal{I}_s)] = \rho\left(\frac{\tau + \kappa}{6} - \frac{1 + \hat{\xi}}{9}\right) + \frac{2(\tau + \kappa)^2}{9} + \frac{\hat{\xi}^2 - \hat{\xi}\tau - \hat{\xi}\kappa - \tau\kappa}{3} - \frac{3\tau + 4\kappa - \hat{\xi} - 3}{18}, \tag{6}$$

*where $\rho = p(\hat{b} = -1|s = 0)$. Suppose that $\hat{\xi} \sim \mathrm{Uni}(\hat{\xi} \mid \hat{\tau} \leq \hat{\xi} \leq \hat{\kappa})$, $\rho \sim \mathrm{Uni}(\rho \mid 0 \leq \rho \leq 1)$, $[\tau, \kappa] \sim \mathrm{Uni}([\tau, \kappa] \mid 0 \leq \tau \leq \kappa \leq 1)$, we have*

$$\mathbb{E}_{\hat{s}, s, \tau, \kappa}[\psi(\mathcal{I}_{\hat{s}}, \mathcal{I}_s)] = 36^{-1}(8\hat{\kappa}^2 + 4\hat{\tau}\hat{\kappa} - 12\hat{\kappa} + 8\hat{\tau}^2 - 8\hat{\tau} + 7),$$
$$\mathbb{E}_{\hat{b}, s, \hat{\xi}, \rho, \tau, \kappa}[\psi(\mathcal{I}_{\hat{b}}, \mathcal{I}_s)] = 16^{-1}(2\hat{\kappa}^2 + 2\hat{\tau}\hat{\kappa} - 3\hat{\kappa} + 2\hat{\tau}^2 - 3\hat{\tau} + 3). \tag{7}$$

$\mathbb{E}_{\hat{s}, s, \tau, \kappa}[\psi(\mathcal{I}_{\hat{s}}, \mathcal{I}_s)] \leq \mathbb{E}_{\hat{b}, s, \hat{\xi}, \rho, \tau, \kappa}[\psi(\mathcal{I}_{\hat{b}}, \mathcal{I}_s)]$ *if and only if* $\left(\hat{\kappa} - \frac{3}{4}\right)^2 + \left(\hat{\tau} - \frac{1}{4}\right)^2 \leq \frac{3}{8}$ *and* $\hat{\tau} \leq \hat{\kappa}$.

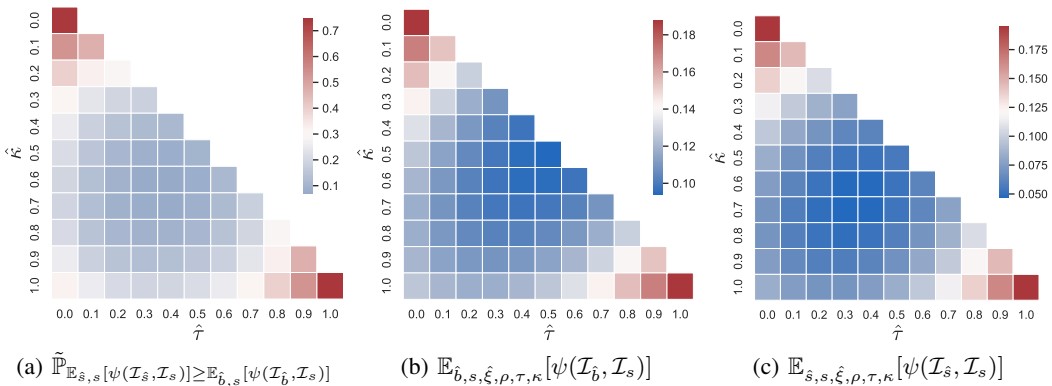

(a) $\tilde{\mathbb{P}}_{\mathbb{E}_{\hat{s},s}[\psi(\mathcal{I}_{\hat{s}},\mathcal{I}_s)]\geq\mathbb{E}_{\hat{b},s}[\psi(\mathcal{I}_{\hat{b}},\mathcal{I}_s)]}$     (b) $\mathbb{E}_{\hat{b},s,\hat{\xi},\rho,\tau,\kappa}[\psi(\mathcal{I}_{\hat{b}},\mathcal{I}_s)]$     (c) $\mathbb{E}_{\hat{s},s,\hat{\xi},\rho,\tau,\kappa}[\psi(\mathcal{I}_{\hat{s}},\mathcal{I}_s)]$

Figure 2: Distributions of the values of $\tilde{\mathbb{P}}_{\mathbb{E}_{\hat{s},s}[\psi(\mathcal{I}_{\hat{s}},\mathcal{I}_s)]\geq\mathbb{E}_{\hat{b},s}[\psi(\mathcal{I}_{\hat{b}},\mathcal{I}_s)]}$, $\mathbb{E}_{\hat{b},s,\hat{\xi},\rho,\tau,\kappa}[\psi(\mathcal{I}_{\hat{b}},\mathcal{I}_s)]$, and $\mathbb{E}_{\hat{s},s,\hat{\xi},\rho,\tau,\kappa}[\psi(\mathcal{I}_{\hat{s}},\mathcal{I}_s)]$ over $[\hat{\tau},\hat{\kappa}]$, where $\mathbb{E}_{\hat{b},s,\hat{\xi},\rho,\tau,\kappa}[\psi(\mathcal{I}_{\hat{b}},\mathcal{I}_s)]$ and $\mathbb{E}_{\hat{s},s,\hat{\xi},\rho,\tau,\kappa}[\psi(\mathcal{I}_{\hat{s}},\mathcal{I}_s)]$ (which are defined in Equation (7)) measures the average EAE of binary labels and ternary labels, respectively; $\tilde{\mathbb{P}}_{\mathbb{E}_{\hat{s},s}[\psi(\mathcal{I}_{\hat{s}},\mathcal{I}_s)]\geq\mathbb{E}_{\hat{b},s}[\psi(\mathcal{I}_{\hat{b}},\mathcal{I}_s)]}$ is defined in Equation (8), which measures the approximate proportion of cases where the ternary label is inferior to the binary label for different $[\tau,\kappa,\hat{\xi},\rho]$.

The detailed proof of Theorem 1 can be found in Appendix. Figure 2 visualizes the results of Theorem 1. Specifically, Figure 2(a) shows the relationship between the predefined parameters $[\hat{\tau},\hat{\kappa}]$ and $\tilde{\mathbb{P}}_{\mathbb{E}_{\hat{s},s}[\psi(\mathcal{I}_{\hat{s}},\mathcal{I}_s)]\geq\mathbb{E}_{\hat{b},s}[\psi(\mathcal{I}_{\hat{b}},\mathcal{I}_s)]}$ which is defined as:

$$\tilde{\mathbb{P}}_{\mathbb{E}_{\hat{s},s}[\psi(\mathcal{I}_{\hat{s}},\mathcal{I}_s)]\geq\mathbb{E}_{\hat{b},s}[\psi(\mathcal{I}_{\hat{b}},\mathcal{I}_s)]} = \frac{1}{|\mathcal{G}|}\sum_{(\tau,\kappa,\hat{\xi},\rho)\in\mathcal{G}}\mathbb{I}(\mathbb{E}_{\hat{s},s}[\psi(\mathcal{I}_{\hat{s}},\mathcal{I}_s)] \geq \mathbb{E}_{\hat{b},s}[\psi(\mathcal{I}_{\hat{b}},\mathcal{I}_s)]),$$
$$\mathcal{G} = \{(\tau,\kappa,\hat{\xi},\rho) : \kappa \in r(1,10^{-3}), \tau \in r(\kappa,10^{-3}), \hat{\xi} \in r(1,10^{-3}), \rho \in r(1,10^{-3})\},$$
$$(8)$$

where $\mathbb{I}(\cdot)$ is an indicator function that outputs 1 if the internal condition is true, and 0 otherwise; $r(u,v)$ outputs an increasing sequence $[0,v,2v,\ldots,u]$. Equation (8) measures the approximate proportion of cases where the ternary label is inferior to the binary label for all possible $[\tau,\kappa,\hat{\xi},\rho]$ in $\mathcal{G}$.

Figure 2(b) and Figure 2(c) show the distributions of the average EAE of the binary label and ternary label, respectively. Obviously, the ternary label outperforms the binary label in most cases. Specifically, the binary label shows superiority only in the extreme cases, i.e., $(\hat{\tau},\hat{\kappa}) \in \{(0,0),(0,0.1),(0.9,1),(1,1)\}$, but $\hat{\tau}$ and $\hat{\kappa}$ basically never take these values in practical applications since both the binary and ternary labels exhibit very high approximation error in these cases. In addition, Theorem 1 also derives that $\mathbb{E}_{\hat{s},s,\tau,\kappa}[\psi(\mathcal{I}_{\hat{s}},\mathcal{I}_s)] \leq \mathbb{E}_{\hat{b},s,\hat{\xi},\rho,\tau,\kappa}[\psi(\mathcal{I}_{\hat{b}},\mathcal{I}_s)]$ if and only if $\left(\hat{\kappa} - \frac{3}{4}\right)^2 + \left(\hat{\tau} - \frac{1}{4}\right)^2 \leq \frac{3}{8}$ and $\hat{\tau} \leq \hat{\kappa}$, which is visualized as the overlapping area of the red and blue regions in Figure 3. It can be seen that the overlapping area is essentially consistent to the blue area in Figure 2(a). Therefore, the ternary label is superior to the binary label w.r.t. approximating the ground-truth label description degrees.

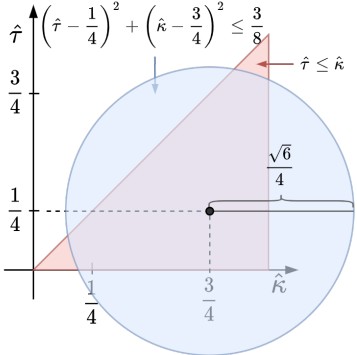

Figure 3: The visualization of $\hat{\tau} \leq \hat{\kappa}$ and $\left(\hat{\kappa} - \frac{3}{4}\right)^2 + \left(\hat{\tau} - \frac{1}{4}\right)^2 \leq \frac{3}{8}$.

## 4   CateMO: Categorical distribution with monotonicity and orderliness

In order to learn label distributions from ternary labels, existing LE methods can be borrowed, the fundamental frameworks of which have been visualized in Figure 4 and illustrated in Section 2. It can be observed that both discriminative and generative LE methods necessitate modeling the relationship between the label description degrees $z$ and the more accessible labels $y$. Therefore,

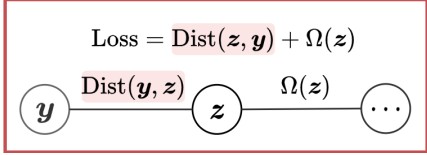
(a) Discriminative label enhancement

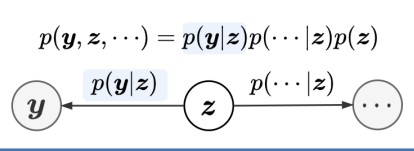
(b) Generative label enhancement

Figure 4: Fundamental frameworks of existing LE methods. $y$ can be either binary labels or multi-label rankings. The learning target of discriminative LE methods can be decomposed as the inconsistency between $y$ and the label description degrees $z$, i.e., $\text{Dist}(z, y)$, and the regularization term of $z$ based on other data, i.e., $\Omega(z)$. The joint distribution of the generation process in generative LE methods can be decomposed as the conditional probability of $y$ given $z$, i.e., $p(y|z)$, and the generative distributions of other observed variables, i.e., $p(\cdots|z)$.

we propose CateMO (Categorical distribution with Monotonicity and Orderliness) to model the conditional probability of ternary label given the label description degree. CateMO can serve as $p(y|z)$ in generative LE methods, and the negative likelihood function of CateMO can be used as $\text{Dist}(z, y)$ in discriminative LE methods, so that most existing LE methods can be employed to address our task by replacing $\text{Dist}(z, s)$ and $p(s|z)$ with CateMO. In the following subsections, we first provide an intuitive discussion of the rules governing the generation of ternary labels and then formalize these rules as the assumptions about the probability monotonicity and orderliness of ternary labels. Furthermore, we derive the parametric mass function for CateMO, which is theoretically guaranteed to maintain the probability monotonicity and orderliness of ternary labels.

## 4.1 Generation rules of ternary labels

On the one hand, we explore how the label description degree affects the probabilities of the label being positive, negative and uncertain. Obviously, a label is more likely to be positive if the description degree of the label is larger; a label is more likely to be negative if the description degree of the label is smaller. Besides, the uncertain label is used to encode situations where the expert is unsure whether the label can describe the instance, which arises from the fact that the probability of the label being positive is close to the probability of the label being negative. Therefore, we believe that a label is more likely to be uncertain if the probabilities of the label being positive and negative are closer. The above intuitions can be formalized as follows:

**Assumption 3 (Probability monotonicity of ternary labels)** *Given any two real values $v_1$ and $v_2$ between 0 and 1, if $v_1 < v_2$, then $p(s = 1|z = v_1) < p(s = 1|z = v_2)$ and $p(s = -1|z = v_1) > p(s = -1|z = v_2)$. If $0 < p(s = 1|z = v_1) - p(s = -1|z = v_1) < p(s = 1|z = v_2) - p(s = -1|z = v_2)$ or $0 < p(s = -1|z = v_1) - p(s = 1|z = v_1) < p(s = -1|z = v_2) - p(s = 1|z = v_2)$, then $p(s = 0|z = v_1) > p(s = 0|z = v_2)$.*

On the other hand, we explore how the label description degree affects the orderliness among the probabilities of the label being positive, negative and uncertain. Obviously, a label is most likely to be negative and least likely to be positive if the label description degree is sufficiently small; a label is most likely to be positive and least likely to be negative if the label description degree is sufficiently large; a label is most likely to be uncertain if the label description degree is moderate. We formalize the intuition as follows:

**Assumption 4 (Probability orderliness of ternary labels)** *There exists two real values $0 \leq v_1 < v_2 \leq 1$, $p(s = 1|z) < p(s = 0|z) < p(s = -1|z)$ holds for any $z < v_1$, $p(s = -1|z) < p(s = 0|z) < p(s = 1|z)$ holds for any $z > v_2$, $\max\{p(s = 1|z), p(s = -1|z)\} < p(s = 0|z)$ holds for any $v_1 < z < v_2$.*

## 4.2 Probability mass function of CateMO distribution

Since a ternary label can take three possible values, i.e., $s \in \{0, \pm 1\}$, we model ternary label by a categorical distribution, which can be formalized as follows:

$$p(s|z) = \text{Categorical}(s \mid [\underline{\varphi}(z), \varphi(z), \overline{\varphi}(z)]), \tag{9}$$

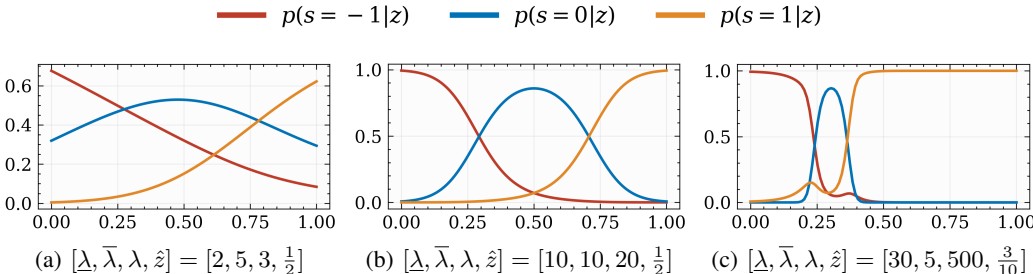

$$\text{(a) } [\underline{\lambda}, \overline{\lambda}, \lambda, \hat{z}] = [2, 5, 3, \tfrac{1}{2}] \qquad \text{(b) } [\underline{\lambda}, \overline{\lambda}, \lambda, \hat{z}] = [10, 10, 20, \tfrac{1}{2}] \qquad \text{(c) } [\underline{\lambda}, \overline{\lambda}, \lambda, \hat{z}] = [30, 5, 500, \tfrac{3}{10}]$$

Figure 5: $p(s|z)$ on different parameters. The horizontal and vertical axes denote the label description degree $z$ and the probability $p(s|z)$ defined by Equation (9) and Equation (10), respectively.

where $\underline{\varphi}(z) = p(s = -1|z)$, $\varphi(z) = p(s = 0|z)$, and $\overline{\varphi}(z) = p(s = 1|z)$ represent the generation principles from the label description degree to the negative, uncertain, and positive labels, respectively. We name $\underline{\varphi}(z)$, $\varphi(z)$, and $\overline{\varphi}(z)$ as ternary generation functions. According to Assumption 3, we preliminarily assume the parametric form of ternary generation functions as follows:

$$\underline{\varphi}(z) = \frac{1}{Z}e^{-\underline{\lambda}z^2}, \quad \varphi(z) = \frac{1}{Z}e^{-\lambda(z-\hat{z})^2}, \quad \overline{\varphi}(z) = \frac{1}{Z}e^{-\overline{\lambda}(z-1)^2}, \tag{10}$$

where $Z = e^{-\underline{\lambda}z^2} + e^{-\lambda(z-\hat{z})^2} + e^{-\overline{\lambda}(z-1)^2}$, $\underline{\lambda} > 0$, $\overline{\lambda} > 0$, $\lambda > 0$ and $0 < \hat{z} < 1$ are parameters. Intuitively, the parameters $(\underline{\lambda}, \overline{\lambda}, \lambda)$ largely governs the precision of CateMO distribution (despite the fact that they are not exactly equal), which is similar to the reciprocal of the temperature coefficient in the softmax layer of a deep neural network. Hence, we refer to the parameters $\underline{\lambda}$, $\overline{\lambda}$, and $\lambda$ as the precision of negative labels, positive labels, and uncertain labels, respectively. Figure 5 visualizes the shape of $p(s|z)$ on different parameters, which can be found that some parameter configurations violate the above three assumptions. For example, Figure 5(a) violates Assumption 3, Figure 5(c) violates Assumption 3 and Assumption 4 at the same time. Therefore, we propose Theorem 2 to ensure that the ternary generation functions defined by Equation (10) satisfy the proposed assumptions about probability monotonicity and orderliness of ternary labels.

**Theorem 2** *Given $\underline{\lambda} > 0$, $\overline{\lambda} > 0$, and $\lambda > 0$, the ternary genertaion functions satisfy Assumption 3 and Assumption 4 if the following conditions hold:*

$$\lambda \neq -\underline{\lambda}\overline{\lambda}(\hat{z}\overline{\lambda} - \hat{z}\underline{\lambda} - \overline{\lambda})^{-1}, \hat{z} = (2\lambda\sqrt{\overline{\lambda}} + 2\lambda\sqrt{\underline{\lambda}})^{-1}(2\lambda\sqrt{\overline{\lambda}} - \underline{\lambda}\sqrt{\overline{\lambda}} + \overline{\lambda}\sqrt{\underline{\lambda}}),$$
$$\max\{(\hat{z} + \hat{z}e^{\overline{\lambda}})^{-1}\overline{\lambda}, ((1 + e^{\underline{\lambda}})(1 - \hat{z}))^{-1}\underline{\lambda}\} < \lambda < \min\{\underline{\lambda}(1 - \hat{z})^{-1}, \overline{\lambda}\hat{z}^{-1}\}. \tag{11}$$

The details of the proof can be found in Appendix. Therefore, the probability mass function of CateMO distribution can be formalized as follows:

$$\text{CateMO}(s = 1 \mid z) = Z^{-1}e^{-\overline{\lambda}(z-1)^2}, \text{CateMO}(s = 0 \mid z) = Z^{-1}e^{-\lambda(z-\hat{z})^2},$$
$$\text{CateMO}(s = -1 \mid z) = Z^{-1}e^{-\underline{\lambda}z^2}, Z = e^{-\underline{\lambda}z^2} + e^{-\lambda(z-\hat{z})^2} + e^{-\overline{\lambda}(z-1)^2},$$
$$\text{s.t. } \hat{z} = (2\lambda\sqrt{\overline{\lambda}} + 2\lambda\sqrt{\underline{\lambda}})^{-1}(2\lambda\sqrt{\overline{\lambda}} - \underline{\lambda}\sqrt{\overline{\lambda}} + \overline{\lambda}\sqrt{\underline{\lambda}}), \lambda \neq -\underline{\lambda}\overline{\lambda}(\hat{z}\overline{\lambda} - \hat{z}\underline{\lambda} - \overline{\lambda})^{-1}, \tag{12}$$
$$\underline{\lambda}, \lambda, \overline{\lambda} > 0, \max\{(\hat{z} + \hat{z}e^{\overline{\lambda}})^{-1}\overline{\lambda}, ((1 + e^{\underline{\lambda}})(1 - \hat{z}))^{-1}\underline{\lambda}\} < \lambda < \min\{\underline{\lambda}(1 - \hat{z})^{-1}, \overline{\lambda}\hat{z}^{-1}\}.$$

Finally, to apply CateMO, we simply replace $\text{Dist}(\boldsymbol{y}, \boldsymbol{z})$ in Figure 4(a) with the negative log-likelihood of CateMO and replace $p(\boldsymbol{y}|\boldsymbol{z})$ in Figure 4(b) with CateMO, which can be formalized as follows:

$$\text{Dist}(\boldsymbol{s}, \boldsymbol{z}) = -\sum_{m=1}^{M} \log \text{CateMO}(s_m \mid z_m), \quad p(\boldsymbol{s}|\boldsymbol{z}) = \prod_{m=1}^{M} \text{CateMO}(s_m \mid z_m). \tag{13}$$

# 5 Experiments

## 5.1 Datasets and evaluation measure

Although there are many LDL datasets, they all lack ground-truth ternary label data. Therefore, we select three real-world LDL datasets (i.e., JAFFE [18], Painting [19], and Music [9]), and manually re-annotate them with both binary labels and the ternary labels. The details of these datasets can be found in Appendix. We use five common LDL metrics to evaluate the algorithm performance, which are Cheb (Chebyshev distance), KL (Kullback-Leibler divergence), Cosine (cosine coefficient) [2], and Rho (Spearman's rho coefficient) [7]. The lower values of Cheb and KL indicate the better performance. The higher values of Cosine, Intersec, and Rho indicate the better performance. We use "↑" and "↓" to denote that the better performance is represented by the higher and lower values of a metric, respectively.

## 5.2 Comparison methods and experimental procedures

**Comparison methods**   Since there is no LE method specifically for ternary labels, we design two approaches to construct the comparison algorithms. On the one hand, we design a data transformation method (which is abbreviated as DT method) to transform the dataset with ternary labels into an extended dataset with binary labels so that any existing LE algorithm can be applied to ternary labels. Specifically, for any instance with uncertain labels, we transform the instance into two instances which set all uncertain labels to positive and negative labels, respectively. For instance, the example $[\boldsymbol{x}, [1, 0, 0]]$ will be transformed into $[\boldsymbol{x}, [1, 1, 1]]$ and $[\boldsymbol{x}, [1, -1, -1]]$, respectively. On the other hand, we replace the loss term $\text{Dist}(\boldsymbol{z}, \boldsymbol{b})$ with MSE $\|(\boldsymbol{s} + 1)/2 - \boldsymbol{z}\|_2^2$, so that most existing LE methods can be used to enhance ternary labels. We abbreviate this method as MSE method. Besides, we select three recently proposed binary label enhancement algorithms: GL [29], LR [6], and MR [12]. The hyperparameter settings follow their respective literature. We combine GL, LR and MR algorithms with DT and MSE methods in pairs to construct six comparison algorithms: GL-DT, GL-MSE, LR-DT, LR-MSE, MR-DT, and MR-MSE. In terms of our proposal, we replace the conditional distribution $p(\boldsymbol{b}|\boldsymbol{z})$ in MR with CateMO and replace $\text{Dist}(\boldsymbol{z}, \boldsymbol{b})$ in GL and LR with the negative log-likelihood function of CateMO, which constructs three algorithms: GL-CateMO, LR-CateMO, and MR-CateMO. We set the parameters $[\underline{\lambda}, \lambda, \overline{\lambda}]$ in CateMO as $[49, 48, 12]$.

**Experimental procedures**   We aim to test the performance of label distribution prediction based on different LE algorithms. Specifically, we use different LE methods to recover training label distributions and use these recovered label distributions to train an LDL model, whose performance on test instances will be reported. LDL-LRR [7] is used as the LDL model in this paper, whose hyperparameters $\lambda$ and $\beta$ are selected from $\{10^{-6}, 10^{-5}, \ldots, 10^{-1}\}$ and $\{10^{-3}, 10^{-2}, \ldots, 10^2\}$ as suggested [7]. We randomly partition the whole dataset (70% for training and 30% for testing), and repeat the above process ten times and report the average and standard deviation of the results.

## 5.3 Results and discussions

Table 1 shows the prediction performance of the comparison algorithms on three datasets. Each result is formatted as "mean±std". In the first column of Table 1, "MR", "LR" and "GL" denote the existing binary LE algorithms, the suffix "-LL" denotes that these algorithms run on binary labels directly, and the suffixes "-DT", "-MSE" and "-CateMO" denote that these algorithms run on ternary labels by the DT method, MSE method and our proposed CateMO distribution, respectively. "Ground-Truth" denotes that LDL-LRR is trained directly on the ground-truth label distributions. In each area separated by dashed lines, **bold** and *italics* denote the 1st and 2nd, respectively. The results of statistical significance test are shown in Appendix. It can be seen that the performance of our proposed CateMO is better than other three comparison algorithms in all cases, and is close to "Ground-Truth". To visualize the advantages of ternary labels over binary labels, we show the relationship between binary labels, ternary labels, and label distributions in terms of annotation time and prediction performance in Figure 6. The horizontal axis denotes the average time (in seconds) spent by an expert in annotating a label for an instance. The vertical axis denotes the average prediction performance calculated from Table 1. It can be seen that ternary labels is superior to binary labels in terms of both prediction performance and annotating cost.

Table 1: Results shown as "mean±std", where **bold** and *italics* denote the 1st and 2nd, respectively.

| Method | Cheb (↓) | KL (↓) | Cosine (↑) | Intersec (↑) | Rho (↑) |
|---|---|---|---|---|---|
| JAFFE | | | | | |
| Ground-Truth | 0.165 ± 0.010 | 0.145 ± 0.015 | 0.897 ± 0.009 | 0.793 ± 0.011 | 0.463 ± 0.051 |
| MR-CateMO | **0.169 ± 0.010** | **0.154 ± 0.015** | **0.892 ± 0.009** | **0.791 ± 0.011** | **0.477 ± 0.051** |
| MR-MSE | 0.183 ± 0.013 | 0.166 ± 0.017 | 0.884 ± 0.010 | 0.774 ± 0.014 | *0.432 ± 0.036* |
| MR-DT | *0.177 ± 0.011* | *0.163 ± 0.012* | *0.886 ± 0.006* | *0.781 ± 0.010* | 0.407 ± 0.033 |
| MR-LL | 0.180 ± 0.010 | 0.171 ± 0.014 | 0.883 ± 0.009 | 0.774 ± 0.008 | 0.383 ± 0.028 |
| LR-CateMO | **0.170 ± 0.012** | **0.156 ± 0.015** | **0.889 ± 0.010** | **0.785 ± 0.010** | **0.443 ± 0.040** |
| LR-MSE | *0.178 ± 0.012* | 0.161 ± 0.017 | 0.884 ± 0.011 | 0.778 ± 0.011 | 0.422 ± 0.045 |
| LR-DT | *0.178 ± 0.012* | 0.161 ± 0.017 | 0.884 ± 0.011 | 0.778 ± 0.011 | *0.424 ± 0.044* |
| LR-LL | 0.201 ± 0.015 | 0.320 ± 0.145 | 0.847 ± 0.023 | 0.738 ± 0.027 | 0.320 ± 0.072 |
| GL-CateMO | **0.174 ± 0.012** | **0.158 ± 0.016** | **0.887 ± 0.011** | **0.782 ± 0.011** | **0.432 ± 0.046** |
| GL-MSE | *0.178 ± 0.012* | *0.162 ± 0.016* | *0.884 ± 0.011* | *0.777 ± 0.011* | *0.422 ± 0.044* |
| GL-DT | 0.206 ± 0.017 | 0.210 ± 0.023 | 0.857 ± 0.013 | 0.745 ± 0.016 | 0.381 ± 0.058 |
| GL-LL | 0.199 ± 0.014 | 0.311 ± 0.156 | 0.852 ± 0.024 | 0.742 ± 0.027 | 0.323 ± 0.078 |
| Painting | | | | | |
| Ground-Truth | 0.252 ± 0.009 | 0.535 ± 0.017 | 0.737 ± 0.007 | 0.605 ± 0.009 | 0.316 ± 0.038 |
| MR-CateMO | **0.262 ± 0.010** | **0.561 ± 0.022** | **0.723 ± 0.008** | **0.593 ± 0.010** | **0.275 ± 0.033** |
| MR-MSE | **0.262 ± 0.011** | 0.564 ± 0.029 | *0.718 ± 0.013* | *0.592 ± 0.013* | 0.255 ± 0.051 |
| MR-DT | 0.263 ± 0.008 | *0.562 ± 0.019* | *0.718 ± 0.009* | *0.592 ± 0.010* | *0.273 ± 0.023* |
| MR-LL | 0.263 ± 0.008 | 0.566 ± 0.019 | 0.715 ± 0.008 | 0.591 ± 0.010 | 0.265 ± 0.047 |
| LR-CateMO | **0.255 ± 0.009** | **0.545 ± 0.021** | **0.731 ± 0.008** | **0.602 ± 0.010** | **0.313 ± 0.044** |
| LR-MSE | 0.266 ± 0.011 | 0.575 ± 0.025 | 0.712 ± 0.011 | 0.587 ± 0.012 | 0.229 ± 0.033 |
| LR-DT | **0.255 ± 0.009** | *0.549 ± 0.025* | 0.730 ± 0.009 | *0.601 ± 0.011* | *0.310 ± 0.042* |
| LR-LL | 0.268 ± 0.014 | 0.617 ± 0.044 | 0.685 ± 0.018 | 0.576 ± 0.014 | 0.195 ± 0.069 |
| GL-CateMO | **0.254 ± 0.010** | **0.542 ± 0.022** | **0.734 ± 0.008** | **0.604 ± 0.011** | **0.316 ± 0.046** |
| GL-MSE | 0.265 ± 0.010 | *0.571 ± 0.027* | 0.715 ± 0.012 | 0.588 ± 0.012 | 0.240 ± 0.044 |
| GL-DT | *0.256 ± 0.011* | 0.590 ± 0.026 | *0.719 ± 0.011* | *0.591 ± 0.012* | *0.287 ± 0.039* |
| GL-LL | 0.260 ± 0.010 | 0.610 ± 0.055 | 0.710 ± 0.008 | 0.588 ± 0.011 | 0.224 ± 0.068 |
| Music | | | | | |
| Ground-Truth | 0.072 ± 0.002 | 0.103 ± 0.006 | 0.925 ± 0.004 | 0.821 ± 0.007 | 0.510 ± 0.025 |
| MR-CateMO | **0.076 ± 0.002** | **0.107 ± 0.007** | **0.921 ± 0.005** | **0.820 ± 0.007** | **0.502 ± 0.027** |
| MR-MSE | **0.076 ± 0.002** | *0.108 ± 0.006* | *0.920 ± 0.004* | *0.817 ± 0.007* | 0.469 ± 0.029 |
| MR-DT | 0.088 ± 0.006 | 0.118 ± 0.009 | 0.910 ± 0.008 | 0.808 ± 0.009 | 0.496 ± 0.034 |
| MR-LL | 0.083 ± 0.004 | 0.115 ± 0.009 | *0.912 ± 0.008* | 0.813 ± 0.009 | *0.499 ± 0.022* |
| LR-CateMO | **0.073 ± 0.002** | **0.103 ± 0.006** | **0.924 ± 0.005** | **0.820 ± 0.007** | **0.506 ± 0.024** |
| LR-MSE | **0.073 ± 0.002** | *0.104 ± 0.006* | **0.924 ± 0.005** | *0.819 ± 0.007* | 0.501 ± 0.027 |
| LR-DT | **0.073 ± 0.002** | *0.104 ± 0.007* | **0.924 ± 0.005** | *0.819 ± 0.007* | *0.504 ± 0.027* |
| LR-LL | 0.081 ± 0.004 | 0.150 ± 0.033 | 0.909 ± 0.008 | 0.806 ± 0.010 | 0.503 ± 0.028 |
| GL-CateMO | **0.073 ± 0.002** | **0.103 ± 0.007** | **0.924 ± 0.005** | **0.821 ± 0.007** | **0.505 ± 0.026** |
| GL-MSE | **0.073 ± 0.002** | *0.104 ± 0.007* | **0.924 ± 0.005** | *0.819 ± 0.007* | *0.501 ± 0.028* |
| GL-DT | 0.084 ± 0.002 | 0.126 ± 0.007 | 0.912 ± 0.004 | 0.809 ± 0.007 | 0.499 ± 0.028 |
| GL-LL | 0.078 ± 0.003 | 0.137 ± 0.009 | 0.906 ± 0.006 | 0.796 ± 0.008 | 0.481 ± 0.033 |

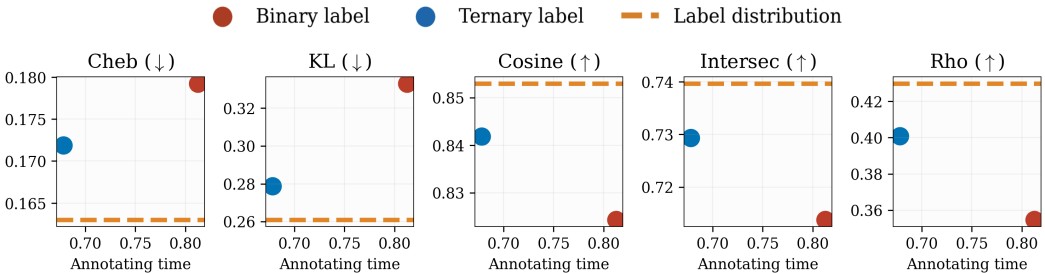

Figure 6: Cost-benefit analysis of different forms of labels. The horizontal and vertical axes denote the average annotating time (in seconds) and performance, respectively.

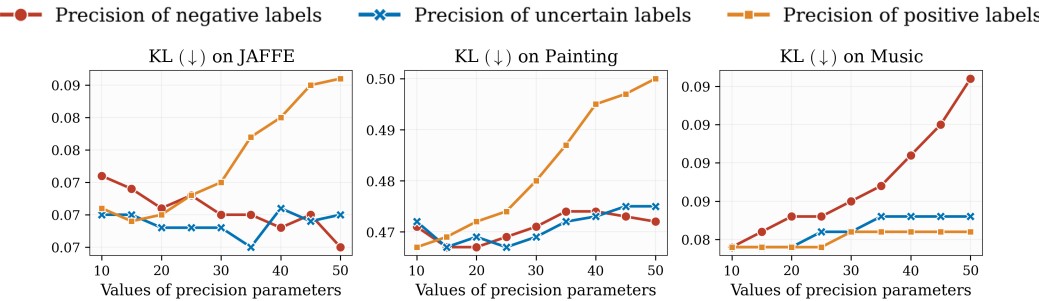

Figure 7: Recovery performance of GL-CateMO with varying precision parameters.

## 5.4 Effect of precision parameters

Figure 7 shows how the precision parameters affect the recovery performance of GL-CateMO on JAFFE, Painting, and Music datasets. The recovery performance is computed by the following two steps. First, we run GL-CateMO algorithm on the dataset with ternary labels to recover the label distributions for instances. Second, we calculate the KL divergence between the recovered label distributions and the ground-truth by the LDL metrics. For a certain precision parameter, the other two precision parameters are set to the values that give rise to the best recovery performance.

## 6 Limitations and conclusion

**Limitations** In this paper, the parameters (i.e., $\underline{\lambda}$, $\lambda$, and $\overline{\lambda}$) of CateMO are pre-fixed. In fact, a more adaptive approach is to collaboratively learn these parameters and other model parameters. However, since the parameters of CateMO satisfy the conditions shown in Theorem 2, in which the parameters are interdependent, it may lead to difficulties in directly exploiting gradient descent optimization methods. Therefore, in future works, we will explore how to appropriately reduce the parameter space of CateMO so that the parameters $\underline{\lambda}$, $\lambda$, and $\overline{\lambda}$ can be decoupled from each others.

**Conclusion** In this paper, we propose to predict label distribution from ternary labels, which reduces both the annotation inaccuracy and cost when contrasted with the traditional binary annotating methods. In the theoretical part, we analyze the approximation error for both ternary and binary labels, which provides a quantitative elucidation of the superior performance of the ternary label. In the methodological part, we propose the CateMO distribution to model the mapping from label description degrees to ternary labels, which is theoretically constructed to maintain the monotonicity and ordinality of the probabilities associated with ternary labels. In the experimental part, extensive experiments demonstrate the effectiveness of our proposal.

## 7 Acknowledgments

This work was partially supported by the National Natural Science Foundation of China (62176123, 62476130), and the Natural Science Foundation of Jiangsu Province (BK20242045).

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

# A Appendix / supplemental material

## A.1 Proof of Theorem 1

Given a label equiprobably being positive, negative, and uncertain, if the label is annotated by a ternary value, then we have the expected approximation error:

$$
\begin{aligned}
\mathbb{E}_{\hat{s},s}[\psi(\mathcal{I}_{\hat{s}},\mathcal{I}_s)] &= \sum_{u\in\{0,\pm 1\}} p(\hat{s}=u, s=u)\psi(\mathcal{I}_{\hat{b}=u},\mathcal{I}_{s=u}) \\
&= \frac{1}{3}\frac{\int_0^{\hat{\tau}}\int_0^{\tau}(z-\hat{z})^2\mathrm{d}z\mathrm{d}\hat{z}}{\hat{\tau}\tau} + \frac{1}{3}\frac{\int_{\hat{\tau}}^{\hat{\kappa}}\int_{\tau}^{\kappa}(z-\hat{z})^2\mathrm{d}z\mathrm{d}\hat{z}}{(\hat{\kappa}-\hat{\tau})(\kappa-\tau)} + \frac{1}{3}\frac{\int_{\hat{\kappa}}^1\int_{\kappa}^1(z-\hat{z})^2\mathrm{d}z\mathrm{d}\hat{z}}{(1-\hat{\kappa})(1-\kappa)} \\
&= \frac{2}{9}(\tau+\kappa)^2 + \frac{2}{9}(\hat{\tau}+\hat{\kappa})^2 - \frac{1}{6}(\hat{\tau}\kappa+\hat{\kappa}\tau) - \frac{1}{3}(\hat{\tau}+\kappa)(\hat{\kappa}+\tau) + \frac{1}{18}(1-\kappa-\hat{\kappa}).
\end{aligned}
\tag{14}
$$

If the label is annotated by a binary value, then we have the expected approximation error:

$$
\begin{aligned}
\mathbb{E}_{\hat{b},s}[\psi(\mathcal{I}_{\hat{b}},\mathcal{I}_s)] &= \sum_{u=\pm 1} p(\hat{b}=u, s=u)\psi(\mathcal{I}_{\hat{b}=u},\mathcal{I}_{s=u}) + p(\hat{b}=u, s=0)\psi(\mathcal{I}_{\hat{b}=u},\mathcal{I}_{s=0}) \\
&= \rho\left(\frac{\tau+\kappa}{6} - \frac{1+\hat{\xi}}{9}\right) + \frac{2(\tau+\kappa)^2}{9} + \frac{\hat{\xi}^2-\hat{\xi}\tau-\hat{\xi}\kappa-\tau\kappa}{3} - \frac{3\tau+4\kappa-\hat{\xi}-3}{18},
\end{aligned}
\tag{15}
$$

where $\rho = p(\hat{b}=-1|s=0)$. Suppose that $\hat{\xi}\sim\mathrm{Uni}(\hat{\xi}\mid\hat{\tau}\le\hat{\xi}\le\hat{\kappa})$, $\rho\sim\mathrm{Uni}(\rho\mid 0\le\rho\le 1)$, $[\tau,\kappa]\sim\mathrm{Uni}([\tau,\kappa]\mid 0\le\tau\le\kappa\le 1)$, we have

$$
\begin{aligned}
\mathbb{E}_{\hat{s},s,\tau,\kappa}[\psi(\mathcal{I}_{\hat{s}},\mathcal{I}_s)] &= \left(\int_0^1\int_0^{\kappa}\mathrm{d}\tau\mathrm{d}\kappa\right)^{-1}\int_0^1\int_0^{\kappa}\mathbb{E}_{\hat{s},s}[\psi(\mathcal{I}_{\hat{s}},\mathcal{I}_s)]\mathrm{d}\tau\mathrm{d}\kappa \\
&= 36^{-1}(8\hat{\kappa}^2 + 4\hat{\tau}\hat{\kappa} - 12\hat{\kappa} + 8\hat{\tau}^2 - 8\hat{\tau} + 7) \\
\mathbb{E}_{\hat{b},s,\hat{\xi},\rho,\tau,\kappa}[\psi(\mathcal{I}_{\hat{b}},\mathcal{I}_s)] &= \frac{1}{\int_0^1\int_0^{\kappa}\int_0^1\int_{\hat{\tau}}^{\hat{\kappa}}\mathrm{d}\hat{\xi}\mathrm{d}\rho\mathrm{d}\tau\mathrm{d}\kappa}\int_0^1\int_0^{\kappa}\int_0^1\int_{\hat{\tau}}^{\hat{\kappa}}\mathbb{E}_{\hat{b},s}[\psi(\mathcal{I}_{\hat{b}},\mathcal{I}_s)]\mathrm{d}\hat{\xi}\mathrm{d}\rho\mathrm{d}\tau\mathrm{d}\kappa \\
&= 16^{-1}(2\hat{\kappa}^2 + 2\hat{\tau}\hat{\kappa} - 3\hat{\kappa} + 2\hat{\tau}^2 - 3\hat{\tau} + 3).
\end{aligned}
\tag{16}
$$

Furthermore,

$$
\begin{aligned}
&\mathbb{E}_{\hat{s},s,\tau,\kappa}[\psi(\mathcal{I}_{\hat{s}},\mathcal{I}_s)] \le \mathbb{E}_{\hat{b},s,\hat{\xi},\rho,\tau,\kappa}[\psi(\mathcal{I}_{\hat{b}},\mathcal{I}_s)] \\
\iff& 8\hat{\kappa}^2 + 4\hat{\tau}\hat{\kappa} - 12\hat{\kappa} + 8\hat{\tau}^2 - 8\hat{\tau} + 7 \le 4\hat{\kappa}^2 + 4\hat{\tau}\hat{\kappa} - 6\hat{\kappa} + 4\hat{\tau}^2 - 6\hat{\tau} + 6 \\
\iff& \hat{\kappa}^2 - \frac{3}{2}\hat{\kappa} + \hat{\tau}^2 - \frac{1}{2}\hat{\tau} + \frac{1}{4} \le 0 \\
\iff& \left(\hat{\kappa}-\frac{3}{4}\right)^2 + \left(\hat{\tau}-\frac{1}{4}\right)^2 \le \frac{3}{8}
\end{aligned}
\tag{17}
$$

## A.2 Proof of Theorem 2

To ensure that Equation (10) satisfies the probability monotonicity of negative and positive labels, we just need to ensure that $\partial\underline{\varphi}(z)/\partial z < 0$ and $\partial\overline{\varphi}(z)/\partial z > 0$. The partial derivative of $\underline{\varphi}(z)$ w.r.t. $z$ can be formalized as:

$$
\frac{\partial\underline{\varphi}(z)}{\partial z} = \left((\lambda z - \hat{z}\lambda - \underline{\lambda}z)e^{v_1} + (\overline{\lambda}z - \overline{\lambda} - \underline{\lambda}z)e^{v_2}\right)\cdot v_3,
\tag{18}
$$

where $v_1, v_2, v_3 > 0$. Since $\overline{\lambda}z - \overline{\lambda} - \underline{\lambda}z < 0$, $(\overline{\lambda}z - \overline{\lambda} - \underline{\lambda}z)e^{v_2} < 0$. Therefore, $\partial\underline{\varphi}(z)/\partial z < 0$ if $\lambda z - \hat{z}\lambda - \underline{\lambda}z < 0$ holds for any $z\in(0,1)$. This inequality can be equivalently transformed into:

$$
\frac{\underline{\lambda}-\lambda}{\hat{z}\lambda} > -1 > -\frac{1}{z} \iff \lambda < \frac{\lambda}{1-\hat{z}}.
\tag{19}
$$

Similarly, the partial derivative of $\overline{\varphi}(z)$ w.r.t. $z$ can be formalized as:

$$
\frac{\partial\overline{\varphi}(z)}{\partial z} = \left((z\lambda - \hat{z}\lambda - \overline{\lambda}z + \overline{\lambda})e^{v_4} + (\overline{\lambda} - \overline{\lambda}z + \underline{\lambda}z)e^{v_5}\right)\cdot v_6,
\tag{20}
$$

where $v_4, v_5, v_6 > 0$. Since $\overline{\lambda} - \overline{\lambda}z + \underline{\lambda}z > 0$, $(\overline{\lambda} - \overline{\lambda}z + \underline{\lambda}z)e^{v_5} > 0$. Therefore, $\partial\overline{\varphi}(z)/\partial z > 0$ if $z\lambda - \hat{z}\lambda - \overline{\lambda}z + \overline{\lambda} > 0$, i.e., $\lambda(z - \hat{z}) > \overline{\lambda}(z - 1)$ holds for any $z \in (0, 1)$. On the one hand, if $\hat{z} < z < 1$, then $\lambda > \overline{\lambda}(z - 1)(z - \hat{z})^{-1}$. On the other hand, if $z < \hat{z}$, then $\lambda < \frac{\overline{\lambda}}{\hat{z}} < \frac{\overline{\lambda}(z-1)}{z-\hat{z}}$.

To ensure that Equation (10) satisfies the probability orderliness, we just need to ensure that equations $\overline{\varphi}(z) = \varphi(z)$ (i.e., $\overline{\lambda}(z - 1)^2 = \lambda(z - \hat{z})^2$) and $\underline{\varphi}(z) = \varphi(z)$ (i.e., $\underline{\lambda}z^2 = \lambda(z - \hat{z})^2$) have only one solution on $z \in (0, 1)$. We denote $g_1(z) = \overline{\lambda}(z - 1)^2 - \lambda(z - \hat{z})^2$. Since $g_1(1) = -\lambda(\hat{z} - 1)^2 < 0$ and $g_1(0) = \overline{\lambda} - \hat{z}^2\lambda > \overline{\lambda} - \hat{z}^2\overline{\lambda}\hat{z}^{-1} = \overline{\lambda}(1 - \hat{z}) > 0$, the quadratic equation $g_1(z) = 0$ has only one solution in $z \in (0, 1)$. Similarly, we denote $g_2(z) = \underline{\lambda}z^2 - \lambda(z - \hat{z})^2$. Since $g_2(1) = 2\hat{z}\lambda - \hat{z}^2\lambda - \lambda + \underline{\lambda} = \underline{\lambda} - (1 - \hat{z})^2\lambda > \underline{\lambda} - (1 - \hat{z})^2\underline{\lambda}(1 - \hat{z})^{-1} = \hat{z}\underline{\lambda} > 0$ and $g_2(0) = -\hat{z}^2\lambda < 0$, the quadratic equation $g_2(z) = 0$ has only one solution in $z \in (0, 1)$.

To ensure that Equation (10) satisfies the probability monotonicity of uncertain label, we just need to ensure that $\partial\varphi(z)/\partial z > 0$ holds for any $0 < z < z_0$ and $\partial\varphi(z)/\partial z < 0$ holds for any $z_0 < z < 1$, where $\overline{\varphi}(z_0) = \underline{\varphi}(z_0)$, i.e., $z_0 = \sqrt{\overline{\lambda}}(\sqrt{\overline{\lambda}} + \sqrt{\underline{\lambda}})^{-1}$ That is, make sure that $z_0$ is the only point of maximum value of $\varphi(z)$. The partial derivative of $\varphi(z)$ w.r.t. $z$ is:

$$\frac{\partial\varphi(z)}{\partial z} = v_1 \cdot \left( (\hat{z}\lambda - \lambda z + \underline{\lambda}z)e^{\overline{\lambda}(z^2+1)} + (\hat{z}\lambda - \lambda z + \overline{\lambda}z - \overline{\lambda})e^{z(2\overline{\lambda}+\underline{\lambda}z)} \right), \qquad (21)$$

where $v_1 > 0$. We aim to make sure that $f(z) = \partial\varphi(z)/\partial z = 0$ has only one solution in $z \in (0, 1)$

$$\underbrace{(-\hat{z}\lambda + \lambda z - \underline{\lambda}z)(\hat{z}\lambda - \lambda z + \overline{\lambda}z - \overline{\lambda})^{-1}}_{f(z)} = \underbrace{\exp(2\overline{\lambda}z + \underline{\lambda}z^2 - \overline{\lambda}z^2 - \overline{\lambda})}_{g(z)}. \qquad (22)$$

We first need to examine whether the denominator of $f(z)$ is zero. Suppose that it is zero, we have $z = (\overline{\lambda} - \hat{z}\lambda)(\overline{\lambda} - \lambda)^{-1}$. Considering that $\overline{\lambda} - \hat{z}\lambda > \overline{\lambda} - \hat{z}\overline{\lambda}\hat{z}^{-1} = 0$, if $\overline{\lambda} > \lambda$, then $z = (\overline{\lambda} - \hat{z}\lambda)(\overline{\lambda} - \lambda)^{-1} > 1$, which contradicts $z \in (0, 1)$. If $\overline{\lambda} < \lambda$, then $z = (\overline{\lambda} - \hat{z}\lambda)(\overline{\lambda} - \lambda)^{-1} < 0$, which also contradicts $z \in (0, 1)$. Besides, it is obvious that the denominator of $f(z)$ is non-zero when $\lambda = \overline{\lambda}$. Therefore, the denominator of $f(z)$ is non-zero. Since whether the partial derivative of $f(z)$ w.r.t. $z$ (i.e., $\partial f(z)/\partial z = (\cdots)^{-2}(\hat{z}\lambda\overline{\lambda} - \hat{z}\lambda\underline{\lambda} - \lambda\overline{\lambda} + \underline{\lambda}\overline{\lambda}))$ is positive or negative has nothing to do with $z$, we ensure the monotonicity of $f(z)$ by the following inequality:

$$\lambda \neq -\underline{\lambda}\overline{\lambda}(\hat{z}\overline{\lambda} - \hat{z}\underline{\lambda} - \overline{\lambda})^{-1}. \qquad (23)$$

Besides, since $\partial g(z)/\partial z = (2\overline{\lambda}(1 - z) + 2\underline{\lambda}z)\exp(\cdot) > 0$, $g(z)$ is increasing. Therefore, to make sure that $\varphi(z)$ has only one maximal point, we just need to ensure that $f(0) > g(0)$ and $f(1) < g(1)$, i.e.,

$$\lambda > \max\left\{ (\hat{z} + \hat{z}\exp(\overline{\lambda}))^{-1}\overline{\lambda}, ((1 + \exp(\underline{\lambda}))(1 - \hat{z}))^{-1}\underline{\lambda} \right\}. \qquad (24)$$

Finally, let the partial derivative of $\varphi(z)$ at $z = z_0$ to be zero, we have

$$\hat{z} = \frac{(\lambda z_0 - \underline{\lambda}z_0)\exp(\overline{\lambda}(z_0^2 + 1)) + (\lambda z_0 - \overline{\lambda}z_0 + \overline{\lambda})\exp\left(z_0(2\overline{\lambda} + \underline{\lambda}z_0)\right)}{\lambda\exp(\overline{\lambda}(z_0^2 + 1)) + \lambda\exp\left(z_0(2\overline{\lambda} + \underline{\lambda}z_0)\right)}. \qquad (25)$$

Considering $z_0 = \frac{\sqrt{\overline{\lambda}}}{\sqrt{\overline{\lambda}}+\sqrt{\underline{\lambda}}}$, we have $\hat{z} = (2\lambda\sqrt{\overline{\lambda}} + 2\lambda\sqrt{\underline{\lambda}})^{-1}(2\lambda\sqrt{\overline{\lambda}} - \underline{\lambda}\sqrt{\overline{\lambda}} + \overline{\lambda}\sqrt{\underline{\lambda}})$. $\qquad\square$

### A.3 Datasets

**JAFFE dataset** The "JAFFE" dataset [18] comprises 213 facial emotion images posed by ten Japanese female models. Concerning the feature data, we employ the feature extraction method suggested in [2] to compress each image into a 243-dimensional feature vector. As for the label data, each image is annotated with scores from $\{1, 2, 3, 4, 5\}$ by 60 individuals, indicating the relevance of the corresponding emotion to the facial image. The average score for each emotion is utilized to represent the intensity of the emotion. Subsequently, we apply min-max normalization to scale the average scores to the interval $(0, 1)$ and normalize the average scores of all emotions into the form of probability distributions to obtain the label distribution data. Besides, three experts annotate the instances in JAFFE with ternary labels and binary labels, and the final ternary labels and binary labels are determined by majority voting.

**Painting dataset**    The "Painting" dataset [19] is created for exploring emotions within abstract art-works. Each image is represented by a 142-dimensional feature vector comprising three components: histogram features of the RGB attributes of the image, histogram features of the HSV attributes of the image, and GLCM (Grey-Level Co-occurrence Matrix) features of the image. Regarding the label data, approximately 230 individuals annotate emotional scores across eight categories: amusement, anger, awe, content, disgust, excitement, fear, sadness. Each image received about 14 annotations. Subsequently, we apply min-max normalization to scale the average emotional scores to the interval $(0, 1)$ and normalize the average scores of all emotions into the form of probability distributions to obtain label distribution data. Besides, three experts annotate the instances in Painting with ternary and binary labels, and the final ternary and binary labels are determined by majority voting.

**Music dataset**    The "Music" dataset is an extension of a music dataset [9] and encompasses 360 popular songs from major music charts across different countries. We employ MFCC (Mel Frequency Cepstral Coefficients) to extract a 5992-dimensional feature vector from each song and utilize PCA (Principal Component Analysis) to compress this vector into a 128-dimensional feature vector. Regarding the label data, participants from the UK, South Korea and Portugal rate their perceived moods (i.e., calm, tense, cheerful, sad, danceable, love, dreamy, electronic, and energy) for a given song on a 4-level scale. Subsequently, we apply min-max normalization to scale the scores to the interval $(0, 1)$, and normalize the average scores of all moods into the form of probability distributions to obtain label distribution data. Besides, three experts annotate the instances in Music with ternary and binary labels, and the final ternary and binary labels are determined by majority voting.

## A.4    Results of statistical significance test

Table 2: The counts of win/tie/loss after comparing CateMO with other comparison algorithms under a pairwise two-tailed $t$-test with $0.05$ significance level. Each entry is formatted as "win/tie/loss".

| Method | Cheb | KL | Cosine | Intersec | Rho |
|---|---|---|---|---|---|
| MSE | 5/4/0 | 5/4/0 | 6/3/0 | 5/4/0 | 7/2/0 |
| DT | 5/4/0 | 6/3/0 | 7/2/0 | 5/4/0 | 6/3/0 |
| LL | 8/1/0 | 9/0/0 | 9/0/0 | 8/1/0 | 9/0/0 |
| Ground-Truth | 0/5/4 | 0/4/5 | 0/4/5 | 0/5/4 | 0/4/5 |

We use pairwise two-tailed $t$-test with $0.05$ significance level to test whether CateMO is statistically superior or inferior to MSE, DT, LL, and ground-truth on three real-world datasets when combined with MR, LR, and GL. The counts of win/tie/loss after comparing CateMO with other comparison algorithms are shown in Table 2.

