# OpenReview forum: "Predicting Label Distribution from Ternary Labels"
_NeurIPS.cc/2024/Conference — NeurIPS 2024 poster_

### Official Review · Reviewer_Lfun · 2024-07-05

**Soundness:** 3
**Presentation:** 3
**Contribution:** 3
**Rating:** 7
**Confidence:** 3

**Summary:**

The paper proposes a more cost-effective approach to label distribution inference, i.e., predicting label distributions from ternary labels. Theoretically, the paper elucidates the superiority of the ternary label by analyzing the error of approximating the ground-truth label description degrees by ternary and binary labels, respectively. Methodologically, the paper proposes a categorical distribution with dimensional monotonicity and orderliness, which is theoretically proven to preserve the monotonicity and ordinality of the probabilities associated with ternary labels, to capture the process of generating ternary labels from label description degrees.

**Strengths:**

1.	Originality: The paper exhibits a high degree of originality. Specifically, by incorporating the philosophies of three-way decision and three-world thinking, the paper provides a novel learning paradigm, i.e., predicting label distribution from ternary labels, to address the trade-off between the accuracy and cost in quantifying label distributions. To the best of my knowledge, this is the first work to consider the philosophies of three-way decision and three-world thinking in the realm of label distribution inference. I am convinced that the combination is logical, as the label distribution encapsulates the polysemy within labels, and the three-way philosophies are established to preserve the uncertainty in decision-making, which means that their objectives are harmoniously aligned.
2.	Quality: The paper is of high quality, as the proposed paradigm and CateMO distribution have been rigorously established through both theoretical foundations and experimental validation. Moreover, the paper also comprises in-depth analysis and discussions of both theoretical and experimental results.
3.	Clarity: The paper is well-articulated, with a coherent and logical structure. Furthermore, the inclusion of clear and relevant visual supports, such as diagrams and figures, effectively enhances the overall understanding of the findings.
4.	Significance: The paper is an impactful contribution to the LDL domain, as it effectively reduces the barriers to entry for implementing LDL algorithms, thereby broadening the spectrum of their applicability.

**Weaknesses:**

1.	There is a lack of introduction to the three-way philosophies. Since most readers may not be familiar with the three-way philosophies, it is necessary for this paper to provide a proper introduction to the three-way philosophies, rather than just presenting their names and references.
2.	Figure 3 and Figure 2 of the paper appear in a wrong order. Figure 3 should be placed after Figure 2.
3.	In Figure 3, the triangle area should be $0\le \hat{\tau}\le \hat{\kappa} \le 1$, instead of $\hat{\tau}\le \hat{\kappa}$, which should be corrected to make the figure more rigorous.

**Questions:**

I have the following two questions. First, how does the proposed CateMO distribution work with existing label enhancement algorithms to infer label distributions from ternary labels? Could you please answer this question with some detailed examples? Second, to offer some intuition for setting the lambda parameters in practical applications, could you please provide a semantic interpretation of the lambda parameters of CateMO distribution?

**Limitations:**

The paper has adequately discussed the limitations and the potential solutions of the proposed method. Besides, I don't believe that the paper has a potential negative social impact.

---

> ### Author Rebuttal · Authors · 2024-08-04
>
> ## Responses to Weaknesses
>
> Thank you for your suggestions. Under the space restriction, we will incorporate some background knowledge about three-way philosophies in the revised version. Besides, we will rearrange the Figure 2 and Figure 3, and correct the mathematical formula in Figure 3.
>
> ## Responses to Questions
>
> Thank you for your questions! Take GLLE as an exmple, the objective function of GLLE is
>
> $$
> \\mathcal L=\\sum\_{n=1}^N \\Vert f(\\boldsymbol x\_n; \\boldsymbol W) - \\boldsymbol y\_n \\Vert\_2^2 + \\Omega(\\boldsymbol W),
> $$
>
> where$f(\\boldsymbol x\_n; \\boldsymbol W)$ is a predictive model with learnable parameters of $\\boldsymbol W$, $\\boldsymbol y\_n$ is a logical label vector, $\\Omega(\\boldsymbol W)$ is the regularization term. If we apply CateMO for GLLE, the objective function will be
>
> $$
> \\mathcal L=\\sum\_{n=1}^N -\\log \\mathrm{CateMO}(\\boldsymbol s\_n|f(\\boldsymbol x\_n; \\boldsymbol W)) + \\Omega(\\boldsymbol W),
> $$
>
> where $\\boldsymbol s\_n$​ is a ternary label vector.
>
> Besides, in terms of how to set the lambda parameters, we provide the following two methods:
>
> - End-to-end Learning. The most straightforward way to determine the parameters $\\underline{\\lambda},\\lambda,\\overline{\\lambda}$ is to co-optimize these parameters with other parameters in the learner in an end-to-end manner. This approach is capable of obtaining parameters adapted to the characteristics of the data distribution, but corresponds to a complex constrained optimization problem that requires a tailored optimization algorithm since the constrains of $\\underline{\\lambda},\\lambda,\\overline{\\lambda}$ are interdependent.
>
> - Pre-annotations Fitting. The parameters can also be estimated from a small number of pre-annotated data pairs $\\{(s\_n,z\_n)\\}\_{n=1}^L$, since $p(s|z)$ possesses a relatively stable morphology across different datasets. Figure 2 in the submitted PDF in global responses shows the distribution of some of our annotation results on the Painting, Music and JAFFE datasets. Specifically, we randomly choose an instance and a label from the JAFFE, Painting, and Music datasets, and asked the experts to simultaneously annotate the instance-label relationship by a ternary value and a description degree value. The above process is repeated six hundred times (two hundred times for each dataset). All annotation results are collected and recorded as $\\mathcal A=\\{(s\_n,z\_n)\\}\_{n=1}^L$. Figure 2 in the submitted PDF in global responses shows the empirical distribution of $p(s|z)$ according to $\\mathcal A$, which indicates that there are no significant differences in the empirical distributions of $p(s|z)$ across datasets. Therefore, we can estimate the parameters of CateMO reliably with a small amount of pre-annotated data pairs. Formally, we obtain the parameters by maximum likelihood estimation, i.e.,
>
> $$
> \\begin{aligned}
> &\\underline{\\lambda}^\\star, {\\lambda}^\\star, \\overline{\\lambda}^\\star \\leftarrow \\arg\\max\_{\\underline{\\lambda}, {\\lambda}, \\overline{\\lambda}}\\quad \\sum\_{(s\_{n}, z\_{n}) \\in \\mathcal A} \\log p(s\_{n} \\vert z\_{n}) \\\\
> \\text{s.t.}\\quad & {\\lambda} < \\min \\{{\\underline{\\lambda}}{(1-\\hat{z})^{-1}}, \\overline{\\lambda} \\hat{z}^{-1}\\},\\\\
> &{\\lambda} \\neq {-\\underline{\\lambda}\\overline{\\lambda}}{(\\hat{z}\\overline{\\lambda} - \\hat{z} \\underline{\\lambda} - \\overline{\\lambda})^{-1}}, \\\\
> &{\\lambda} > \\max\\left\\{\\left( \\hat{z} + \\hat{z} \\exp(\\overline{\\lambda}) \\right)^{-1}\\overline{\\lambda}, ((1+\\exp(\\underline{\\lambda})) (1-\\hat{z}))^{-1} \\underline{\\lambda} \\right\\} , \\\\
> & \\hat{z} = \\left( 2{\\lambda} \\sqrt{\\overline{\\lambda}} + 2{\\lambda} \\sqrt{\\underline{\\lambda}} \\right)^{-1} \\left(2{\\lambda} \\sqrt{\\overline{\\lambda}} - \\underline{\\lambda} \\sqrt{\\overline{\\lambda}} + \\overline{\\lambda} \\sqrt{\\underline{\\lambda}}\\right).
> \\end{aligned}
> $$
>
>
> Besides, in order to provide users with a more intuitive understanding of CateMO's parameters, we show the shape of CateMO with varying $\\underline{\\lambda},\\lambda,\\overline{\\lambda}$ in Figure 3 in the submitted PDF in global responses, which visualizes how the parameters $\\underline{\\lambda},\\lambda,\\overline{\\lambda}$ detemine CateMO.

---

### Official Review · Reviewer_7vWN · 2024-07-11

**Soundness:** 4
**Presentation:** 4
**Contribution:** 4
**Rating:** 8
**Confidence:** 5

**Summary:**

The authors propose a novel multi-label annotation scheme, transitioning from the traditional binary annotation to a ternary annotation. They have validated this new annotation scheme through both theoretical analysis and experimental verification.

**Strengths:**

1. The authors clearly articulate the problem they aim to solve, and the illustrative diagrams effectively demonstrate the problem's setup.

2. The proposed scheme is validated from both theoretical and experimental perspectives.

**Weaknesses:**

The field of multi-label learning has developed various learning models to address different limitations of the label space, such as weakly supervised learning, partial multi-label learning, multi-label learning with class-conditional noise, and weakly semi-supervised multi-label learning. Research in these areas is well-established. The relationship between the "uncertain relevant" concept mentioned in the paper and existing work needs further clarification.

**Questions:**

1. The proposed approach of quantifying the advantages of ternary labels over binary labels through approximation error analysis is commendable. However, I am curious about the impact of the proportion of "uncertain relevant labels" in practical scenarios. In other words, if the proportion of "uncertain relevant labels" is excessively high within the overall labeling, does the proposed ternary annotation scheme still offer a statistically significant advantage over the traditional binary annotation scheme?

2. While the authors validate the superiority of the ternary annotation scheme through experiments, the fairness of the existing experimental setup in terms of method comparison needs further clarification. As the paper suggests, there is no established LE method for ternary labels, and the experimental model proposed by the authors is new. It is recommended that the authors discuss the fairness of the comparative experiments in greater detail.

**Limitations:**

the authors adequately addressed the limitations

---

> ### Author Rebuttal · Authors · 2024-08-04
>
> ## Responses to Weakness
>
> We really appreciate your valuable comments. Essentially, partial multi-labels, weakly-supervised multi-labels, semi-supervised multi-labels, and multi-labels with noise are essentially weaker versions of binary labels (i.e., multi-labels) because they either contain noise (i.e., incorrectly supervision data) or miss the correct supervision data. In contrast, ternary labels represent an enhanced version of binary labels, as ternary labels not only provide definitely relevant labels and definitely irrelevant labels but also leverage the option of uncertain label to identify the labels that are more prone to generating noise. Besides, the detailed illustration about the relationship between this paper and weakly-supervised multi-label learning is shown in Responses to Weakness (1) for Reviewer 3ktH. Under the space restriction, we will incorporate a discussion about the relationships between our paper and other similar paradigms in the revised version.
>
> ## Responses to Questions
>
> Thank you for your insightful questions! In terms of the cases where the number of uncertain labels is excessively large, we give the following proposition:
>
> ***Proposition***: *Given an uncertain label, i.e., $p(s=0)=1$, we denote the expected approximation errors produced by annotating the label with a ternary value and a binary value as $\\psi\_s$ and $\\psi\_b$, respectively. Suppose that* $\\hat\\xi\\sim \\rm{Uni}(\\hat \\xi\\mid \\hat\\tau \\le \\hat \\xi \\le \\hat\\kappa)$, $\\rho\\sim \\rm{Uni}(\\rho\\mid 0\\le\\rho\\le 1)$, $[\\tau,\\kappa]\\sim \\rm{Uni}([\\tau,\\kappa]\\mid 0\\le \\tau\\le \\kappa\\le 1)$*, we have*
>
> $$
> \\mathbb{E}\_{\\hat\\xi,\\rho,\\tau,\\kappa}[\\psi\_s - \\psi\_b] = \\frac29(\\hat\\tau^2+\\hat\\kappa^2+\\hat\\tau\\hat\\kappa) - \\frac13(\\hat\\tau+\\hat\\kappa)+\\frac{1}{12}.
> $$
>
> *Furthermore, if $[\\hat\\tau,\\hat\\kappa]\\sim \\rm{Uni}([\\hat\\tau,\\hat\\kappa]\\mid 0\\le \\hat\\tau\\le \\hat\\kappa\\le 1)$, we have*
>
> $$
> \\begin{aligned}
> &\\mathbb P\_{[\\hat\\tau,\\hat\\kappa]\\sim \\rm{Uni}([\\hat\\tau,\\hat\\kappa]\\mid 0\\le \\hat\\tau\\le \\hat\\kappa\\le 1)}\\left[\\mathbb{E}\_{\\hat\\xi,\\rho,\\tau,\\kappa}[\\psi\_s - \\psi\_b]\\le 0\\right] \\\\
> =&  \\int\_{\\frac29(\\hat\\tau^2+\\hat\\kappa^2+\\hat\\tau\\hat\\kappa) - \\frac13(\\hat\\tau+\\hat\\kappa)+\\frac{1}{12}\\le 0, 0\\le \\hat\\tau\\le \\hat\\kappa\\le 1} \\rm{d} \\hat\\tau \\rm{d}\\hat\\kappa  \\left(\\int\_{0\\le \\hat\\tau\\le \\hat\\kappa\\le 1}\\rm{d} \\hat\\tau \\rm{d}\\hat\\kappa\\right)^{-1}\\\\
> =&\\frac{6+6\\sqrt 3 + \\sqrt 3\\pi}{24}\\approx 0.91.
> \\end{aligned}
> $$
>
> The above proposition reveals the advantage of ternary labels over binary labels when all labels are uncertain labels, i.e., $p(s=0)=1$. According to the above proposition, the expected approximation error of binary annotating exceeds that of ternary annotating when $\\frac29(\\hat\\tau^2+\\hat\\kappa^2+\\hat\\tau\\hat\\kappa) - \\frac13(\\hat\\tau+\\hat\\kappa)+\\frac{1}{12}\\le 0$. Furthermore, if we equiprobably set $\\hat\\tau$ and $\\hat\\kappa$ to any two values in the interval $0\\le \\hat\\tau\\le \\hat\\kappa\\le 1$, then the probability that the ternary annotating possesses lower expected approximation error is 91% approximately. Besides, in terms of the comparative experiments, we will clarify the fairness of each comparative experiments in greater detail under the space restriction.

---

### Official Review · Reviewer_3ktH · 2024-07-11

**Soundness:** 1
**Presentation:** 3
**Contribution:** 2
**Rating:** 3
**Confidence:** 5

**Summary:**

In this manuscript, the authors explore how to learn the unknown label distribution from given ternary labels (0, 1, and -1). The main contributions of this work include: (1) a new label distribution prediction method is designed for handling ternary labels; (2) the theoretical analysis on the error of approximating the GT label description degrees by ternary and binary labels has been done.

**Strengths:**

(1)	The theoretical analysis on the error of approximating the GT label description degrees by ternary and binary labels is detailed.

(2)	The proposed MR-CateMO can deliver better results than that of baseline methods on three datasets.

**Weaknesses:**

(1)	Actually, some similar work has been done on “predicting label distribution from ternary labels” by researchers. For example, the problem definition of this manuscript is same to [17]. Although [17] has been discussed in the related work part, the authors did not point out that this work has made effort to address the same problem. This is quite strange! From this perspective, the novelty of this task is limited. Moreover, some claims of the authors maybe wrong. For example, “Since there is no LE method for ternary labels, we design…”.

(2)	Experiments are insufficient, and some existing label enhancement algorithms should be compared, e.g., [a], [b], [17]. Moreover, according to the suggestion of [17], there are many datasets can be used for this task, but the authors of this manuscript only evaluate their approach on three datasets.

[a] Fusion Label Enhancement for Multi-Label Learning. IJCAI, 2022.
[b] Label enhancement via manifold approximation and projection with graph convolutional network. Pattern Recognition, 2024.

**Questions:**

(1) I think that the task of “Prediction label distribution from ternary labels” is a branch of weakly supervised learning. However, there is no discussion between them in this manuscript.

(2) Actually, prediction unknown label distribution or label enhancement has been employed to deal with some other machine learning tasks, e.g., multi-label learning, partial label learning, feature selection and so on. But, the authors do not mention the possible extension of the proposed model to these or other scenarios.

(3) The advantages and disadvantages between the presented algorithm and existing discriminative/generative label enhancement ones should be discussed.

(4) The objective function of the proposed method is not shown. I think it is important for readers to clearly understand that how the method learn the unknown label distributions from ternary labels. By the way, I do not think the unknown label description degree is equal to the probability of an instance belonging to the positive label or negative label, though they are usually positively correlated with each other. So, I wonder that the proposed model how to enrich the supervised information in the label space and generate the label description degree.

(5) From Table 1, I find that the results of different approaches on some metrics are closed. Therefore, the statistical significance analysis should be done, which can show whether the accuracy differences are significant or not.

(6) The parameter sensitivity analysis of the three hyper-parameters in the proposed model is missed.

**Limitations:**

I find that LDL-LRR is used as the LDL model in the testing phase. This two-stage strategy is a limitation of its application in real-world scenarios. I encourage the authors to talk about the possible solution on designing a one-stage extension model based on the proposed method.

---

> ### Author Rebuttal · Authors · 2024-08-04
>
> ## Responses to Weakness (1)
>
> (1) Responses to the concern about "Weakly Supervised Multi-Label Learning via Label Enhancement".
>
> We greatly appreciate the constructive feedback you have provided on our paper. __*It should be highly emphasised that the ternary labels in our paper and the WSML (i.e., weakly-supervised multi-labels) are inherently different despite sharing the same representation ($-1$, $0$, $1$).*__ Specifically, they differ in the following aspects.
>
> | Summary of Difference | Weakly-Supervised Multi-Labels, WSML | Ternary Labels |
> | -- | -- | -- |
> | They differ in the origin of the label with value of $0$, i.e., the missing label in WSML and the uncertain label in ternary labels. | The missing label in WSML stems from its "absence", i.e., the association between the label and the corresponding instance is undocumented or unannotated rather than undeterminable. | The uncertain label in ternary labels stems from its "uncertainty", i.e., it is difficult for experts to definitively determine whether the label is relevant to the corresponding instance. |
> | They differ in the range of the underlying label description degree. | The description degree of the missing label to the corresponding instance take values in the interval $[0,1]$, i.e., $[0,\\tau)\\cup [\\tau,\\kappa]\\cup (\\kappa,1]$, since the missing label may be a relevant label, an irrelevant label or an uncertain label. | The description degree of the uncertain label to the corresponding instance take values in the interval $[\\tau,\\kappa]$. |
> | They differ in the informativeness of the supervision data. | WSML is less informative than binary label (a.k.a. multi-label) due to the loss of information pertaining to some of the relevant and irrelevant labels. | Ternary label is more informative than binary label since it additionally encodes the cases where the label is neither definitely relevant nor definitely irrelenvant to the instance. |
>
>
> Admittedly, we acknowledge that our paper may not illustrate this aspect thoroughly. We pledge to improve this section in accordance with your valuable suggestions.
>
> (2) Responses to why there is no LE (i.e., Label Enhancement) method for ternary labels.
>
> As illustrated in the above table, WSML and ternary labels differ in the origin of the label with value of $0$​. Therefore, we believe that it is unjustified to apply the algorithms of WSML directly for ternary labels. Nonetheless, we acknowledge your suggestions and we will further improve the claims.
>
> ## Responses to Weakness (2)
>
> (1) Responses to the suggestion that some existing LE methods should be compared, e.g., [a], [b], [17].
>
> Our proposed CateMO distribution, serving as a plug-and-play component that enables most standard binary LE algorithms to be applied to ternary labels, rather than an algorithm that outperforms existing binary LE algorithms. Therefore, our proposed CateMO is not comparable to the papers [a], [b], [17]. Nevertheless, we will discuss the relationships between the papers [a], [b], [17] and our paper appropriately since these papers are related to our work.
>
> (2) Responses to the concern about datasets.
>
> The datasets used for the experiments in the paper [17] are not applicable to the research context of this paper because those datasets lack ternary labels. However, ternary labels cannot be obtained by setting the binary labels to uncertain labels directly since the uncertain labels must be the labels that are really difficult to determine whether they are relevant to the instance.
>
> ## Responses to Question (1)
>
> The fundamental differences between WSML and ternary labels have already been illustrated in "Responses to Weakness (1)". We will comprehensively disscuss their relationships in the revision.
>
> ## Responses to Question (2)
>
> Thank you for your suggestions. Our paper primarily concentrates on the quantitative and qualitative relationships between ternary labels and label description degrees. Unfortunately, due to page limitation, possible extensions of our proposal were not elaborated upon. Under the space restriction, we will endeavor to incorporate a discussion of these extensions in the revised version.
>
> ## Responses to Question (3)
>
> In our paper, we have not proposed an "algorithm". As illustrated in Responses to Weakness (2), our proposed CateMO is a probability distribution, a plugin that is able to work with most discriminative/generative LE algorithms. Therefore, it is unnecessary to discuss the advantages/disadvantages of our proposed CateMO compared to the existing discriminative/generative LE algorithms.
>
> ## Responses to Question (4)
>
> There is no objective function to show in our paper since we have not proposed a “learning algorithm” in our paper. The mathematical details of our proposed CateMO are shown in Eq. (12), and Eq. (13) shows how CateMO works in conjunction with existing LE algorithms. Besides, in terms of how to enrich the supervision information, various LE algorithms have their own logic to enrich the supervision or to generate the label description degrees, e.g., GLLE uses the smoothness assumption, and LELR uses the potential ranking relationships within binary labels. Our proposed CateMO merely models the probabilistic relationship between the label description degree and the ternary label, which is theoretically guaranteed to satisfy the proposed basic assumptions.
>
> ## Responses to Question (5)
>
> The results of statistical significance test have been shown in the Table 2 of Appendix A.4 of our paper.
>
> ## Responses to Question (6)
>
> Thank you for your suggestion. We additionally perform experiments to analyze the parameter sensitivity. The results are shown in Figure 1 in the PDF of global response, which will be added to the revised version as space permits. More detailed discussion about the parameters $\\underline \\lambda,\\lambda,\\overline \\lambda$ can be found in the "Responses to Weakness (1) and Question (2)" section for Reviewer uVMp.

---

> ### Author Response · Authors · 2024-08-12
>
> Dear Reviewer:
>
> We would like to kindly inquire whether our responses have adequately addressed the concerns you raised during the review process. Besides, we are also seeking to ascertain if there are any new question that you would like us to clarify.

---

### Official Review · Reviewer_uVMp · 2024-07-11

**Soundness:** 3
**Presentation:** 3
**Contribution:** 3
**Rating:** 6
**Confidence:** 4

**Summary:**

The authors of this paper propose to predict label distribution from ternary labels, i.e., “0” indicating “uncertain relevant”, “1” indicating “definitely relevant” and “-1” indicating “definitely irrelevant”. Besides, they also provide theoretical analysis to show that the ternary label outperforms the binary label in most cases.

**Strengths:**

1. The proposed method is intuitive and straightforward.
2. This paper proves that the ternary label is superior to the binary label in terms of expected approximation error in most cases.

**Weaknesses:**

1. The proposed Categorical distribution with Monotonicity and Orderliness (CateMO) contains four parameters, and parameter-sensitive analyses are absent.
2. It is not explicitly shown whether the label distribution degrees generated by the CateMO are close to the real label distribution.

**Questions:**

1. Why do you use the Eq.(10) as the ternary generation functions?
2. Are there any guidelines for selecting the parameters in Eq.(10)?
3. Is the proposed CateMO method effective in generating accurate label distribution?

**Limitations:**

Yes, the authors have adequately addressed the limitations.

---

> ### Author Rebuttal · Authors · 2024-08-04
>
> ## Responses to Weakness (1) and Question (2)
>
> We are very grateful for your precious suggestions. We additionally perform experiments to analyze the parameter sensitivity. The results are shown in Figure 1 in the submitted PDF in global responses, which will be added to the revised version if space permits.
>
> Actually, there are only three parameters in CateMO, since the paremeter $\\hat z$ can be determined by the other parameters according to Eq. (12). In terms of how to determine the parameters $\\underline{\\lambda},\\lambda,\\overline{\\lambda}$ in Eq. (10), we offer the following two approaches.
>
> - End-to-end Learning. The most straightforward way to determine the parameters $\\underline{\\lambda},\\lambda,\\overline{\\lambda}$ is to co-optimize these parameters with other parameters in the learner in an end-to-end manner. This approach is capable of obtaining parameters adapted to the characteristics of the data distribution, but corresponds to a complex constrained optimization problem that requires a tailored optimization algorithm since the constrains of $\\underline{\\lambda},\\lambda,\\overline{\\lambda}$ are interdependent.
>
> - Pre-annotations Fitting. The parameters can also be estimated from a small number of pre-annotated data pairs $\\{(s\_n,z\_n)\\}\_{n=1}^L$, since $p(s|z)$ possesses a relatively stable morphology across different datasets. Figure 2 in the submitted PDF in global responses shows the distribution of some of our annotation results on the Painting, Music and JAFFE datasets. Specifically, we randomly choose an instance and a label from the JAFFE, Painting, and Music datasets, and asked the experts to simultaneously annotate the instance-label relationship by a ternary value and a description degree value. The above process is repeated six hundred times (two hundred times for each dataset). All annotation results are collected and recorded as $\\mathcal A=\\{(s\_n,z\_n)\\}\_{n=1}^L$. Figure 2 in the submitted PDF in global responses shows the empirical distribution of $p(s|z)$ according to $\\mathcal A$, which indicates that there are no significant differences in the empirical distributions of $p(s|z)$ across datasets. Therefore, we can estimate the parameters of CateMO reliably with a small amount of pre-annotated data pairs. Formally, we obtain the parameters by maximum likelihood estimation, i.e.,
>
> $$
> \\begin{aligned}
> &\\underline{\\lambda}^\\star, {\\lambda}^\\star, \\overline{\\lambda}^\\star \\leftarrow \\arg\\max\_{\\underline{\\lambda}, {\\lambda}, \\overline{\\lambda}}\\quad \\sum\_{(s\_{n}, z\_{n}) \\in \\mathcal A} \\log p(s\_{n} \\vert z\_{n}) \\\\
> \\text{s.t.}\\quad & {\\lambda} < \\min \\{{\\underline{\\lambda}}{(1-\\hat{z})^{-1}}, \\overline{\\lambda} \\hat{z}^{-1}\\},\\\\
> &{\\lambda} \\neq {-\\underline{\\lambda}\\overline{\\lambda}}{(\\hat{z}\\overline{\\lambda} - \\hat{z} \\underline{\\lambda} - \\overline{\\lambda})^{-1}}, \\\\
> &{\\lambda} > \\max\\left\\{\\left( \\hat{z} + \\hat{z} \\exp(\\overline{\\lambda}) \\right)^{-1}\\overline{\\lambda}, ((1+\\exp(\\underline{\\lambda})) (1-\\hat{z}))^{-1} \\underline{\\lambda} \\right\\} , \\\\
> & \\hat{z} = \\left( 2{\\lambda} \\sqrt{\\overline{\\lambda}} + 2{\\lambda} \\sqrt{\\underline{\\lambda}} \\right)^{-1} \\left(2{\\lambda} \\sqrt{\\overline{\\lambda}} - \\underline{\\lambda} \\sqrt{\\overline{\\lambda}} + \\overline{\\lambda} \\sqrt{\\underline{\\lambda}}\\right).
> \\end{aligned}
> $$
>
>
> Besides, in order to provide users with a more intuitive understanding of CateMO's parameters, we show the shape of CateMO with varying $\\underline{\\lambda},\\lambda,\\overline{\\lambda}$ in Figure 3 in the submitted PDF in global responses, which visualizes how the parameters $\\underline{\\lambda},\\lambda,\\overline{\\lambda}$ detemine CateMO.
>
> ## Responses to Weakness (2) and Question (3)
>
> Thank you for your questions. In Figure 3 in the submitted PDF in global responses, we show, across three datasets, the proximity of the label distributions recovered by CateMO to the ground-truth label distributions. From the experimental results, it can be seen that the label enhancement algorithm based on CateMO achieves significant advantages over others.
>
> ## Responses to Question (1)
>
> The proposed ternary generation function can be expressed as
>
> $$
> \\mathrm{softmax}(-\\underline{\\lambda} z^2,-\\lambda (z-\\hat z)^2, -\\overline\\lambda (z-1)^2),
> $$
>
> where $\\mathrm{softmax}(a,b,c)=[e^a/(e^a+e^b+e^c),e^b/(e^a+e^b+e^c),e^c/(e^a+e^b+e^c)]$. It can be seen that the proposed ternary generation function is essentially a softmax function. The reason why we use the softmax function as the basic form is that the softmax function is most commonly used for modelling probability mass functions of discrete variables and offers great simplicity in differential calculus.
>
> The motivation for the components $-z^2,-(z-\\hat z)^2,-(z-1)^2$ is that the ternary generation functions are expected to maintain the following properties:
>
> - $s$​ is more likely to be $-1$​ when $z$​ is close to $0$​.
> - $s$ is more likely to be $1$ when $z$ is close to $1$.
> - $s$ is more likely to be $0$ when $z$ is far away from both $0$ and $1$​.
>
> Finally, the parameters $\\underline \\lambda,\\lambda,\\overline \\lambda$ are used to control the strength of each component in the softmax function, similar to the precision parameter in a Gaussian distribution.

---

> > ### Comment · Reviewer_uVMp · 2024-08-12
> >
> > Thanks for your responses. I have thoroughly reviewed the comments from other reviewers and the corresponding responses. I find that most of my concerns have been addressed by the authors. Currently, I have decided to retain my score.

---

### Author Rebuttal · Authors · 2024-08-04

We sincerely value the time and thoughtfulness each reviewer has dedicated to enhancing the quality of our paper. We have carefully considered each comment and ensured that each point is addressed. Attached please find a PDF file containing the relevant figures mentioned in the responses for each reviewer. Should any reviewer have further questions or require additional clarification, we are readily available and would be more than happy to assist.

---

### Author Response · Authors · 2024-08-12

Dear Reviewers:

The deadline imminent. Are there any new concerns about our paper or response? If so, please do not hesitate to communicate with us, and we will respond promptly.

---

### Decision · Program_Chairs · 2024-09-25

**Decision:**

Accept (poster)

**Comment:**

The paper proposes a solution that generates label distributions from ternary labels -1/0/1, where “0” indicates “uncertain relevant”, “1” indicates “definitely relevant” and “-1” indicates “definitely irrelevant”.  Three reviewers provided positive reviews. The proposed method is intuitive and straightforward, and the paper is well-articulated, with a coherent and logical structure. This paper proves that the ternary label is superior to the binary label in terms of expected approximation error in most cases.

However, one reviewer raised the concern about the novelty of this paper. During the discussion period, concerns about the novelty of the issue, the limited methodology and the incomplete experiments were raised:

1. The formulation of "predicting label distribution from ternary labels " seems like a label enhancement (LE) process on -1/0/1 labels, i.e., a modified version of label enhancement. The introduction also shows that the issue arises from LE and the proposed methods just directly utilized existing LE methods with a minor change in the conditional probability.

2. This paper provided a solution for generating label distribution from 1/0/1 labeled data but failed to apply the solution to any learning tasks. Many top conference papers on LE applied the LE technique, i.e., generating label distribution to solve many learning tasks（e.g., label distribution learning, multi-label learning, partial label learning) in the experiments to illustrate the improvement of the LE technique for machine learning.

3. The theoretical results cannot prove that the ternary label outperforms the binary label in all cases. The theoretical analysis in this paper just calculated the expected approximation error between generated label distributions and the ground-truth label distributions. However, the theoretical results about the enhancement of the classifier's generalization ability by generating label distribution from ternary labels might be more solid and meaningful than the current theoretical analysis.

4. In the methodology, the proposed method merely adapts existing label enhancement techniques, introducing only a minor alteration to the conditional probability of the input labels. This modification contributes minimally to the field of label enhancement or label distribution learning.

After discussion with the reviewers and the SAC, we are inclined to accept this paper, however, the authors should revise all the aforementioned limitations in the camera-ready paper. Here is the suggestion on how they might address the limitations:

1. In the introduction section, it is necessary to clearly articulate the similarities and differences between this paper and the concept of label enhancement.

2.  In the experimental section, the authors can adopt the proposed method for some learning tasks（e.g., label distribution learning, multi-label learning, partial label learning) to illustrate the improvement of their method for machine learning.

3. The theoretical result shows that the ternary label cannot outperform the binary label in some cases. The authors should fix them. The theoretical results about the enhancement of the classifier's generalization ability by generating label distribution from ternary labels would be appreciated.

4. In the methodology, the proposed method merely adapts existing label enhancement techniques, introducing only a minor alteration to the conditional probability of the input labels. More technique contributions should be provided.